# Contrasting ice formation in Arctic clouds: surface coupled vs decoupled clouds

Hannes J. Griesche[1], Kevin Ohneiser[1], Patric Seifert[1], Martin Radenz[1], Ronny Engelmann[1], and Albert Ansmann[1]

[1]Leibniz Institute for Tropospheric Research (TROPOS), Leipzig, Germany

**Correspondence:** Hannes Jascha Griesche (griesche@tropos.de)

**Abstract.** In the Arctic summer of 2017 (1 June to 16 July) measurements with the OCEANET-atmosphere facility were performed during the Polarstern cruise PS106. OCEANET comprises amongst other instruments the multiwavelength polarization lidar Polly$^{XT}$_OCEANET and was for PS106 complemented with a vertically pointed 35-GHz cloud radar. In the scope of the presented study, the influence of cloud height and surface coupling on the probability of clouds to contain and form ice is investigated. Polarimetric lidar data was used for the detection of the cloud base and the identification of the thermodynamic phase. Both radar, and lidar were used to detect cloud top. Radiosonde data was used to derive the thermodynamic structure of the atmosphere and the clouds. The analyzed data set shows a significant impact of the surface coupling state on the probability of ice formation. Surface-coupled clouds were identified by a quasi-constant potential temperature profile from the surface up to liquid layer base. Within the same minimum cloud temperature range ice-containing clouds have been observed more frequently than surface-decoupled clouds by a factor of up to 6 (temperature intervals between -7.5 and -5°C, 164 vs. 27 analyzed intervals of 30 minutes). The frequency of occurrence of surface-coupled ice-containing clouds was found to be 2-3 times higher (e.g. 82% vs. 35% between -7.5 and -5°C). These findings provide evidence that above -10°C heterogeneous ice formation in Arctic mixed-phase clouds occurs by a factor of 2-6 more often when the cloud layer is coupled to the surface. In turn, for minimum cloud temperatures below -15°C, the frequency of ice-containing clouds for coupled and decoupled conditions approached the respective curve for the Central-European site of Leipzig, Germany (51°N, 12°E). This cooperates the hypothesis that the free-tropospheric ice nucleating particle (INP) reservoir over the Arctic is controlled by continental aerosol. Two sensitivity studies, using also the cloud radar for detection of ice particles, and applying a modified coupling-state detection, both confirmed the findings, albeit with a lower magnitude. Possible explanations for the observations are discussed by considering recent in-situ measurements of INP in the Arctic, of which much higher concentrations were found in the surface-coupled atmosphere in close vicinity to the ice shore.

# 1 Introduction

The Arctic climate is known to change much faster compared to other regions on Earth, which is referred to as Arctic amplification (Serreze and Francis, 2006). The surface temperature anomaly of the Arctic for the year 2013 with respect to the mean
of $1970-1999$ is $2-3\,\mathrm{K}$ warmer compared to the mid-latitudes (Francis and Skific, 2015). As is also pointed out by Francis and Skific (2015), this differential heating will likely have consequences for the mid-latitudinal circulation, leading to reduced zonal winds and consequently more-steady weather periods with accompanied larger regional risk of severe droughts or wet periods.

Given the possible widespread consequences of Arctic amplification, it is essential to understand the physical processes
leading to this rapid change. A number of different atmospheric and marine processes are currently discussed as potential sources for Arctic amplification. Nevertheless, a clear causal chain could not be established (Serreze and Barry, 2011; Pithan and Mauritsen, 2014; Stuecker et al., 2018) because the quantitative contribution of the single processes involved as well as their autocorrelation could not yet be determined to date. Besides the evident role of sea ice loss in the warming process, a key role in Arctic amplification is attributed to clouds (Vavrus, 2004). Kay and L'Ecuyer (2013) obtained a climatology of
Arctic clouds and radiation conditions for the first decade of the 21st century. They highlight the importance of clouds in the Arctic climate system but they also note that both conditions - presence as well as absence of clouds - can contribute to Arctic amplification, depending on the season and the sea ice conditions. In 2016 the German transregio initiative ArctiC Amplification: Climate Relevant Atmospheric and SurfaCe Processes, and Feedback Mechanisms (AC)[3] was established to further investigate the reasons and consequences of Arctic amplification (Wendisch et al., 2019).

With respect to the microphysical properties of Arctic mixed-phase clouds, it is evident that an accurate representation of the latter in atmospheric models is important in order to understand and accurately simulate Arctic climate (Engström et al., 2014). Especially their longevity puts high demands on the research community. Big efforts were put into establishing model frameworks that are capable of simulating such cloud systems (Fridlind et al., 2007; Klein et al., 2009; van Diedenhoven et al., 2009; Morrison et al., 2011; Neggers et al., 2019), even in the case of single-layer clouds of low complexity. The different
processes involved to form and sustain supercooled liquid or mixed-phase clouds were thoroughly discussed by Morrison et al. (2012). Large-scale advection of water vapor is considered to be the prerequisite for formation and persistence of Arctic stratus decks, especially over closed ice surfaces.

In the marginal sea ice zone, the transition zone between closed ice surface and open sea, the importance of surface sources of heat and moisture in promoting cloud processes relative to in-cloud or advective sources is, however, uncertain (Harrington
and Olsson, 2001; Shupe et al., 2013). The subsequent microphysical evolution of the Arctic stratiform cloud decks is subject to the availability of cloud condensation nuclei (CCN) and ice nucleating particles (INP) (Stephens, 2005; Fridlind et al., 2007; Kalesse et al., 2016). At temperatures above -38°C only heterogeneous ice formation takes place, which requires the availability of such INP (Hoose and Möhler, 2012), with INP from different origins starting to be active at different temperatures. Mineral dust, e.g., starts getting active below a temperature of about -15°C (Hoose and Möhler, 2012) while sea spray aerosol INP
have been found to be already active at -5°C (DeMott et al., 2016). INP from biological origin are assumed to be one of the

most active ones at low to moderate supercooling (Schnell and Vali, 1976; Szyrmer and Zawadzki, 1997; Murray et al., 2012; O'Sullivan et al., 2018).

Many studies report that respective INP reservoirs for Arctic clouds are mainly provided by means of long-range transport from lower latitudes (Morrison et al., 2012). An increasing number of studies, however, suggest that also local aerosol sources can provide significant numbers of CCN and INP that stem from marine processes (Bigg, 1996; DeMott et al., 2016; Hartmann et al., 2020) or even from ship emissions or industry (Creamean et al., 2018; Thomson et al., 2018). This suggests that also Arctic clouds are subject to anthropogenic climate change (Lohmann, 2017). Wex et al. (2019) found an annual cycle in INP number concentration (INPC) at four different-land based stations in the Arctic with the largest INPC in summer. Hartmann et al. (2020) analyzed filter measurements from the Arctic airborne campaign PARMACMiP 2018, which was performed in late winter in the vicinity of the Villum Research Station, Greenland (81°N, 16°W) above the Arctic Ocean and Fram Strait. They found the highest INPC during low level flights above open leads and polynyas. Heat sensitivity of the sampled INP as well as high freezing onset hint towards biogenic origin. Low flight altitudes, a large number of open leads in the vicinity of the aircraft flight track, and detected sea salt in the aerosol samples suggest that these INP rather originate from local marine sources than long range transport.

One way of evaluating the relationship between temperature, aerosol conditions, and the efficiency of heterogeneous ice formation is the utilization of remote-sensing observations. From combined observations of cloud radar, lidar, and microwave radiometer, the vertical structure and microphysical composition of clouds and precipitation over a specific site can be obtained (Illingworth et al., 2007; Shupe, 2007). For vertically and optically thin cloud layers also the application of single systems such as polarization lidar can be used to obtain the required information (Sassen, 2005; Ansmann et al., 2009). Thermodynamic properties of the atmosphere are provided by soundings or model data. Numerous studies provide evidence that the occurrence and efficiency of heterogeneous ice formation at ambient conditions depends strongly on both, temperature (Shupe, 2011; Zhang et al., 2017) as well as the type and quantity of the aerosol burden at cloud level (Sassen, 2005; Seifert et al., 2010; Kanitz et al., 2011; Zhang et al., 2018).

With respect to the humidity conditions, such studies are even further constrained under the presence of layers of supercooled liquid water which is the case for the majority of cloud layers with top temperatures above -25°C (Ansmann et al., 2008; de Boer et al., 2011; Westbrook and Illingworth, 2011). Changing aerosol conditions in the Arctic have thus the potential to modify the general occurrence of heterogeneous ice formation and the cloud microphysics. This puts a definite requirement to advance the current understanding of the heterogeneous ice formation in Arctic clouds. Norgren et al. (2018) show that aerosols might be responsible for the reduction in the cloud ice content in low-level Arctic mixed-phase clouds. They found that mixed-phase clouds present in a clean aerosol state have ice water content by a factor of 1.22 to 1.63 higher at cloud base than similar clouds in cases with higher aerosol loading. Jouan et al. (2014) hypothesized that emissions of $SO_2$ may reduce the ice nucleating properties of INP through acidification, resulting in a smaller concentration of larger ice crystals that leads to an increase in precipitation.

Clouds in general are highly variable in their occurrence and structure (Stephens, 2005). A common feature of Arctic clouds is associated with the frequent occurrence of multi-layer temperature and humidity inversions which lead to the formation of

temporally stable multi-level mixed-phase cloud decks (Shupe et al., 2011; Morrison et al., 2012; Verlinde et al., 2013). Yet, Arctic clouds also form commonly under stable conditions with turbulence initiation after persisting for several hours due to radiative cooling (Silber et al., 2020). The cloud layers are of complex macro- and microphysical structure and frequently occur at heights close to the ground which are not easily trackable. Liu et al. (2017) pointed out that space-borne remote-sensing

techniques fail to detect 25 to 40% of the clouds below 500 m height and also underestimate the fraction of mixed-phase and ice clouds between the surface and 1000 m height. In turn, ground-based profiling studies from the Arctic, which rely on lidar and radar observations, usually provide reasonable data only at heights above 100 − 150 m above ground, as is the case for the 35-GHz cloud radar (KAZR) of the U.S. Department of Energy (DOE) Atmospheric Radiation Measurement (ARM) program at the NSA (North Slope of Alaska) site in Utqiaġvik (formerly known as Barrow), USA. Cloud processes that take place at

lower heights can thus not thoroughly be characterized (Griesche et al., 2020).

Even though indications are given that local aerosol sources may play a role for heterogeneous ice formation, none of the studies available so far investigated any potential effects of the surface coupling state of Arctic clouds on the frequency and efficiency of ice formation. Investigation of potential effects of the surface coupling were so far restricted to bulk properties such as ice water path (IWP) or liquid water path (LWP), without referencing clearly to any relations between ice formation

and temperature, or even aerosol conditions. Shupe et al. (2013) found only moderate differences in coupled versus decoupled clouds. They report that clouds which are thermodynamically linked with the surface tend to show colder temperature profiles within the cloud and slightly weaker in-cloud turbulence, yet often have higher LWP and IWP, for which they suggest as a reason the additional moisture supply from below. Qiu et al. (2015) studied the occurrence of Arctic mixed-phase clouds in relation to the presence and strength of humidity and temperature inversions but they did not provide any information about

the overall frequency of ice formation in the different coupling states. Similar to Qiu et al. (2015), Qiu et al. (2018) used the opportunity to study the influence of both surface conditions and different air masses on thermodynamic variables and on the properties of Arctic mixed-phase clouds. Due to the coastal location of the Utqiaġvik site in northern Alaska where the data set for their study was obtained, marine air masses are transported by northerly winds, while more continental air masses are transported by southerly winds. Furthermore, the Arctic mixed-phase cloud occurrence frequency was found to have a positive

correlation with relative humidity with respect to ice and a negative relationship with stability. But this study investigated mixed-phase cloud properties only. The efficiency of ice formation was not investigated. Sotiropoulou et al. (2014) provide a detailed study of the properties of coupled and decoupled Arctic clouds but found with respect to the thermodynamic phase partitioning, that the IWP and LWP as well as their ratio of coupled and decoupled clouds are similar. Gierens et al. (2020) studied surface coupling effects on mixed-phase clouds based on a two year data set from ground-based remote-sensing in

Ny-Ålesund, Svalbard. They found a seasonal cycle of the coupling-state, with most coupled clouds observed during summer. The LWP in coupled clouds was roughly 40% higher compared to decoupled clouds, but only minor differences have been found in IWP. Their findings are effected by the surrounding orography of the measurement site. Glacier outflows tend to be decoupled, while for clouds transported from the open sea towards Ny-Ålesund coupling was most common. The open sea west of Svalbard also might act as a local humidity and heat source. Furthermore, models have their difficulties to accurately

reproduce heterogeneous ice formation in clouds. Nomokonova et al. (2019) reported in agreement with Sandvik et al. (2007)

that single-layer mixed-phase clouds tend to be underestimated in models compared to results from the synergy of different ground-based instruments. Without considering any surface coupling effects in their study, they found in a temperature regime between -20 to -5°C a lower occurrence of mixed-phase clouds at the expense of pure ice clouds.

Given the indication that aerosols play a dominating role in the heterogeneous ice formation process and that Arctic clouds are frequently occurring either coupled or decoupled to the surface and corresponding local aerosol sources, it should be investigated if the characteristics of heterogeneous ice formation processes differ between coupled and decoupled clouds. The goal of this study is motivated by the need for an accurate characterization of the near-surface cloud properties and the prevalent indications that the microphysical and dynamical structure of surface-coupled Arctic clouds differs from those of decoupled clouds (Shupe et al., 2013; Qiu et al., 2015, 2018). The work is based on a comprehensive data set of remote-sensing instruments and atmospheric soundings from an 8-week cruise of the research vessel Polarstern into the marginal ice zone between Greenland and Svalbard in May-July 2017 that was collected in the frame of (AC)[3]. By splitting this data set into low- and high-altitude cloud layers as well as into coupled and decoupled clouds, an investigation of cloud macro- and microphysical properties will be possible separately for free-tropospheric clouds, not depending on regional effects and further aerosol input, and of surface-coupled clouds, being linked to local phenomena and aerosols in the Arctic region.

The article is structured as follows. Section 2 will focus on the instrumentation and methodology used to analyze the data from the ship cruise in the Arctic summer. In Section 3 an overview of the observations will be provided and statistical evaluation of the analyzed measurements of the Arctic clouds will be presented. A detailed discussion of the findings and the methodology is given in Sect. 4. Section 5 summarizes and concludes this study.

## 2   Instrumentation and Methodology

From 1 June until 16 July 2017 cruises PS106.1 and PS106.2 (PS106 in general, see Fig. 1) of the German research vessel (RV) Polarstern took place with the goal to conduct measurements in the marginal sea ice zone north and northeast of Svalbard (Macke and Flores, 2018). Cruise PS106.1 comprised the "Physical feedbacks of Arctic boundary layer, Sea ice, Cloud and AerosoL (PASCAL)" icebreaker expedition and ice floe camp, as well as the "Arctic CLoud Observations Using airborne measurements during polar Day (ACLOUD)" aircraft campaign (Wendisch et al., 2019). PASCAL as well as ACLOUD were dedicated to the investigation of processes related to Arctic amplification. During the full period of PS106, continuous remote-sensing of aerosols and clouds was performed with the OCEANET-atmosphere platform aboard Polarstern (Griesche et al., 2020). The suite of instruments of OCEANET operated during PS106 is listed in Table 1. Of specific interest for the underlying study are the motion-stabilized vertically pointing 35-GHz cloud Doppler radar Mira-35 (Görsdorf et al., 2015; Griesche et al., 2020), the multiwavelength Raman polarization lidar Polly$^{XT}$_OCEANET (hereafter referred to as Polly$^{XT}$; Engelmann et al., 2016) and the microwave radiometer (MWR) HATPRO (Rose et al., 2005). Auxiliary data for the study was obtained from the regularly performed atmospheric soundings of type Vaisala RS92-SGP which are available every 6th hour (UTC) of the day for the entire cruise.

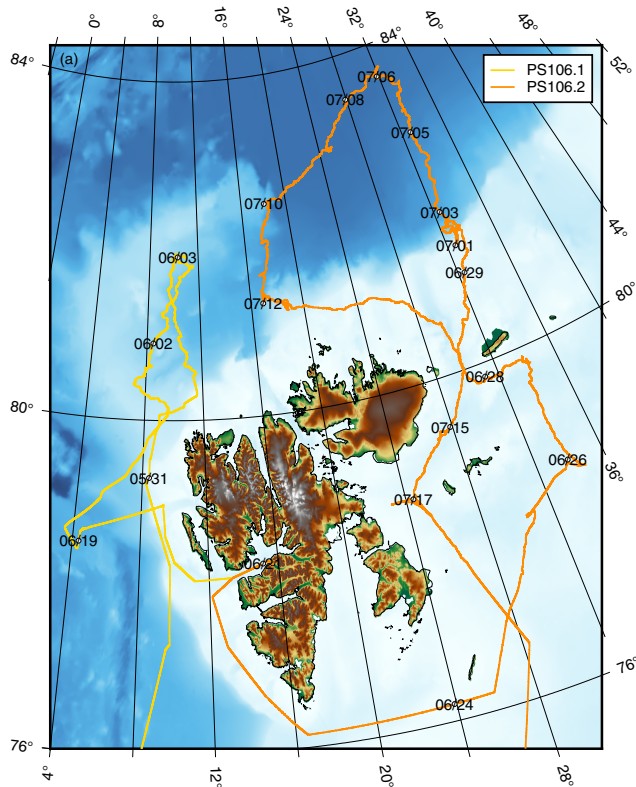

**Figure 1.** Cruise track of the PS106 campaign (with dates annotated at the track). The first leg (PS106.1 / PASCAL) is shown in yellow, the second leg (PS106.2) in orange. Figure taken from Griesche et al. (2020).

The set of instruments deployed for this study is used to obtain information about cloud vertical extent, atmospheric thermo-dynamic state, phase partitioning, and ice and liquid microphysical properties. As mentioned in the introduction, the goal of this study is to investigate the phase partitioning of Arctic cloud systems with respect to their surface coupling. It was thus the objective to obtain similar statistics as presented before by, e.g., Ansmann et al. (2009), Seifert et al. (2010, 2015), and Kanitz et al. (2011). Kanitz et al. (2011) showed that the relationship of spatially and vertically distinct ice-containing cloud layers and cloud top temperature varies strongly by region on Earth. For the current study, however, the cloud classification procedure that was applied by Kanitz et al. (2011) or similar ones such as of Seifert et al. (2010, 2015), was extended in such a way to account for the long-lasting nature of Arctic cloud systems which frequently prevented the classification of distinct, vertically and temporally separated cloud layers. Hence, we now present the applied method for this data set.

**Table 1.** Instrumentation used in the frame of this study.

| Platform | | | |
|---|---|---|---|
| Instrument | Type | Atmospheric parameters | Resolution |
| **OCEANET** | | | |
| Polly$^{XT}$ | Multiwavelength Raman polarization lidar, pointed 5° off- zenith | Particle backscatter and extinction coefficient; linear depolarization ratio; water vapour mixing ratio | 7.5 m; 30 s |
| Mira-35 | 35-GHz (Ka-band) motion stabilized, vertically pointing cloud radar | Vertical structure, boundaries and vertical-velocity dynamics of clouds and precipitation; contributes to cloud liquid water and ice water profiles | 30 m; 3 s |
| HATPRO-G2 | 14-channel microwave radiometer | Estimated profiles of temperature and humidity; integrated water vapor and liquid water path | 100 – 1000 m; 1 s |
| **Polarstern Meteorology** | | | |
| RS92-SGP | Radiosonde | Atmospheric pressure, temperature, humidity, wind vector | 1 s |

## 2.1 Ice-containing cloud analysis

The applied procedure to identify and characterize individual clouds is illustrated in Fig. 2. Initially, the data set is split into
time intervals of 30 minutes. The subsequent analysis is based on the respective 30 minute data portions of lidar attenuated
backscatter coefficient and volume depolarization ratio at 532 nm wavelength, of radar reflectivity factor Z as well as of the
temporally closest radiosonde ascent.

The base of the liquid-dominated layer (hereafter referred to as liquid layer) was determined using the attenuated backscatter
coefficient as observed with the 532 nm near-field channel of Polly$^{XT}$. Data from this channel is required in order to be able
to determine the base of even the lowest clouds which can be as low as 50 m above the surface (Griesche et al., 2020). The
minimum base height of each liquid layer detected in a 30 minute interval was identified using the attenuation approach. Each
7.5 m range gate of each 30 s lidar profile within the 30 min interval was checked for a decrease of the 532 nm attenuated
backscatter coefficient by at least a factor of 10 within 250 m vertical distance. Such a signal drop can only be caused by liquid
clouds. The liquid layer base of each 30 s lidar profile is then set to the height where the gradient for the first time reached
25% of the maximum gradient within these 250 m. This approach is similar to the one used within Cloudnet (Illingworth
et al., 2007), but by omitting a distinct threshold of the attenuated backscatter coefficient, as overlap effects in the near-field
prevented a thorough calibration of the latter at heights below 120 m (Engelmann et al., 2016). The lowest liquid layer base in
the 30 minute period was then assigned as the liquid layer base height of the respective interval.

If applicable the cloud top height was determined using the cloud radar observations. The highest, consecutive cloud radar
pixel connected temporally (within the 30 min interval) and spatially to the liquid layer base was set as cloud top (the minimum
signal detection threshold level of the radar to detect a cloud pixel is 5 times the signal-to-noise ratio detected in the co-channel

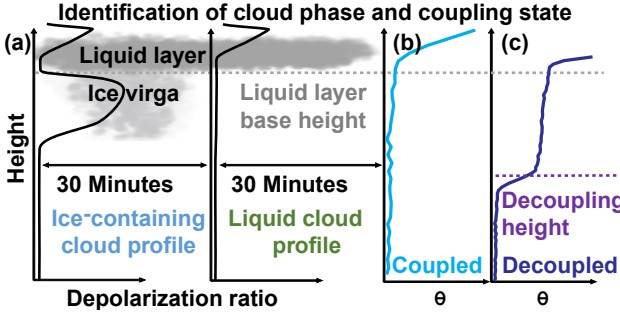

**Figure 2.** Sketch of the applied method: panel (a) shows two example profiles of the lidar volume depolarization ratio. In the first profile an ice virga is present below a liquid layer. Due to the ice below the liquid layer (high depolarization) the first 30 minutes are classified as an ice-containing cloud profile. Without ice falling out of the cloud only the liquid layer is present (low depolarization below the liquid layer). In the presence of ice the liquid layer base height is characterized by a strong decrease in depolarization as the signal is now dominated by the return of the liquid droplets. Within the liquid layer the depolarization increases again due to multiple scattering. In panel (b) and (c) two profiles of $\theta$ are depicted, with (b) illustrating a coupled cloud and (c) a decoupled cloud. In (c) additionally the decoupling height is marked.

of the cloud radar). In the case of clouds below the lowest detection limit of the cloud radar the cloud top was determined using the lidar observations. Finally, the 30 minute interval was screened for the presence of ice virga below the liquid layer base by means of the Polly[XT] 532 nm volume depolarization ratio. Based on theoretical considerations a general quantification of the lidar 532 nm volume depolarization ratio for ice detection was made with the conclusion that a volume depolarization ratio of >0.03 can be interpreted as ice occurrence (note that the molecular depolarization for the Polly[XT] 532 nm channel is <0.01; Engelmann et al., 2016).

In Fig. 2 (a) a simplified profile of the depolarization ratio is shown in case of a cloud present above the lidar. Ice particles show an enhanced depolarization ratio due to their non-spherical crystal shape as can be seen in the first profile in the ice virga below the liquid layer. The signal from the liquid layer is dominated by the return of the cloud droplets (even though a few ice crystals would be present), as the lidar is more sensitive to the larger total surface area of the more numerous cloud droplets. The liquid droplets produce a very low depolarization ratio due to their rather spherical shape. Within the liquid layer, however, the depolarization ratio increases linearly due to multiple scattering (Jimenez et al., 2020). The following classification was applied in the frame of the inspection of each individual 30 minute period: A depolarization ratio close to 0 accompanied by a strong lidar backscatter indicates the presence of spherical liquid-water droplets. If the depolarization ratio below the liquid layer is high (likely due to ice crystals falling out of the cloud) the cloud is classified as an ice-containing cloud (15-16 UTC and 17-18 UTC in Fig. 3). Otherwise, it is classified as a liquid cloud (16-17 UTC and 18-21 UTC in Fig. 3).

The minimum cloud temperature is an important parameter in the sense of this study, as INP efficiency increases by about an order of magnitude every 5 K (DeMott et al., 2015). Hence, the probability of ice production is highest in the cloud where the temperature is lowest. Especially in the Arctic, where temperature inversions at cloud top are frequent, special care must be taken in determining the actual cloud minimum temperature. Using the temperature at the identified cloud top height

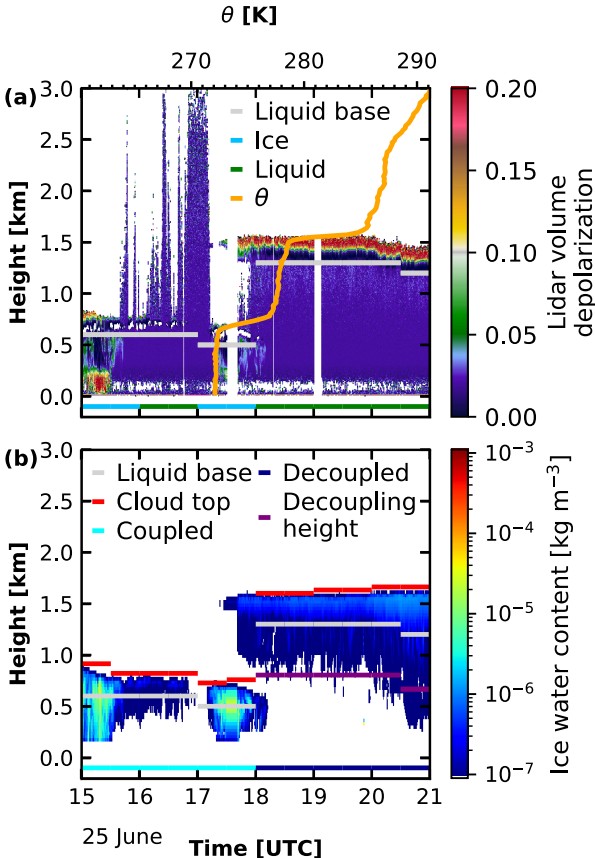

**Figure 3.** An example of the applied method on 25 June 2017 between 15 – 21 UTC. In (a) the lidar volume depolarization is shown. Marked are also flags for liquid layer base height (grey) and cloud phase (ice: blue, liquid: green). Additionally the $\theta$ profile for the sounding launched at 17:15 UTC (orange) is plotted. In (b) the cloud radar reflectivity is depicted, together with information on the liquid base height (grey), cloud top height (red), the coupling state (cyan: coupled, dark blue: decoupled) and (if applicable) the decoupling height (purple).

might produce a bias toward positive temperatures because the likelihood to select a temperature from within or above the temperature inversion is high when using the highest detected cloud pixel. The approach was thus to assign the minimum temperature between cloud base and top as the minimum cloud temperature.

Cloud layers, that might be affected by overlaying hydrometeors were filtered from the data set. Precipitation from the upper layer may act as ice nuclei in the lower one (Vassel et al., 2019). The analyzed data set is corrected for this possible seeding effect based on cloud radar observations, which can detect clouds up to the tropopause, depending on their size and the concurrent atmospheric attenuation. Since the targeted clouds contain rather low amounts of liquid water, the attributed attenuation effects on the cloud radar can likely be neglected. Cloud layers that are vertically closer than 1000 m to the subjacent

cloud are thought to be able to influence the lower one and thus these periods have been excluded from the analysis.

### 2.1.1 Surface coupling state

The surface coupling state of the cloud is derived from the thermodynamic profiles of the radiosondes. Following Gierens et al. (2020), who introduced a simplified version of the coupling algorithm from Sotiropoulou et al. (2014), we examined the profile of the potential temperature $\theta$ starting at liquid layer base down to the surface. If the difference between the cumulative mean of $\theta$ and $\theta$ exceeded 0.5 K the cloud is considered as decoupled (Fig 2 (c)) and this height in taken as the decoupling height (marked in purple in Fig 2 (c) and Fig. 3 (b)). A quasi-constant $\theta$-profile on the other hand identifies coupled clouds (Fig 2 (b)). In Fig. 3 (a) the base of the clouds between $15-17$ UTC were too low to be decoupled ($500-600$ m). The $\theta$-profile was nearly constant until cloud base. From 17 UTC on, however, the liquid base height is significantly higher ($1200-1300$ m) and due to the increase in $\theta$ at roughly 700 m these clouds are defined as decoupled.

## 3 Results

### 3.1 Campaign overview

The investigated period covers 1520 analyzed intervals of 30 minutes. In 88% of these periods a cloud was identified and roughly 57% of the investigated clouds were identified as ice containing clouds. Approximately 62% of the analyzed clouds were coupled to the surface whereas 38% were decoupled. 64% of the surface-coupled clouds were defined as ice-containing clouds but only 47% of the decoupled clouds.

### 3.2 Influence of surface coupling

Following Kanitz et al. (2011) we analyzed the fraction of ice-containing clouds with respect to all observed clouds in different intervals of minimum cloud temperature in the range of heterogeneous freezing, starting at -40 up to 0°C. Figure 4 (a) shows the fraction of ice-containing clouds as a function of minimum cloud temperature for the Arctic (green) in contrast to findings from Leipzig (orange; Kanitz et al., 2011). Below a minimum cloud temperature of -25°C most clouds from both data sets contained ice. At higher temperatures (minimum cloud temperature >-10°C) above Leipzig on the other hand usually little to no ice-containing clouds were found. For the Arctic we found a different pattern in this temperature regime. Temperatures slightly below freezing are already sufficient for ice production: 35 to 70% of the investigated clouds with minimum cloud temperatures above -15°C showed signals of ice. As higher minimum cloud temperatures are usually associated with lower cloud heights, in a next step we analyzed the data set in terms of liquid layer base height. In all analyzed temperature or liquid layer base height intervals, liquid containing clouds were identified.

Figure 4 (b) represents the fraction of ice-containing clouds as a function of liquid layer base heights between 50 and 4000 m. In general there is a tendency of increasing fraction of ice-containing clouds with increasing base height. An increase of liquid layer base height may be associated with an increase of cloud top height, depending on the cloud vertical extent. A higher cloud top height in turn goes typically along with a decrease of minimum cloud temperature. This leads to a higher probability of ice formation, as more aerosols can act as possible INP. The fraction of ice-containing clouds for liquid layer base heights

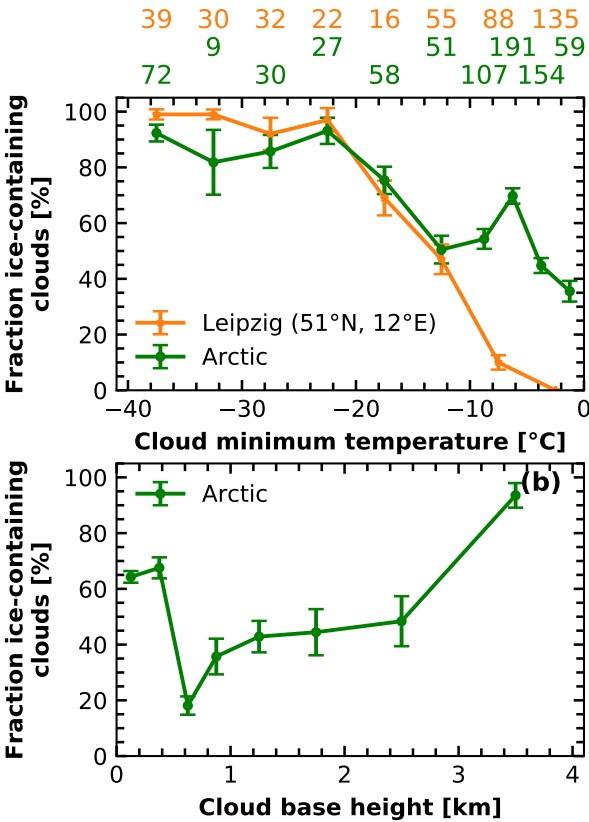

**Figure 4.** (a) Fraction of ice-containing clouds as a function of minimum cloud temperature within the heterogeneous ice nucleation regime. Data of all clouds of the Arctic 2017 field campaign is plotted in green. In orange the results for Leipzig from Fig. 3 in Kanitz et al. (2011) is shown. The error bars indicate the statistical uncertainty as in Seifert et al. (2010). Temperature intervals increase with decreasing minimum cloud temperature due to decreased number of data ($\Delta T$ = 2.5°C above -10°C and 5°C below -10°C). The numbers on top of the plot show the number of data for each temperature interval and the data points have been placed in the middle of the respective investigated interval. (b) Fraction of ice-containing clouds for different liquid layer base height intervals. Base height intervals increase with increasing liquid layer base height due to decreased number of data ($\Delta h$ = 0.25 km below 1 km, 0.5 km between 1 and 2 km and 1 km above 2 km).

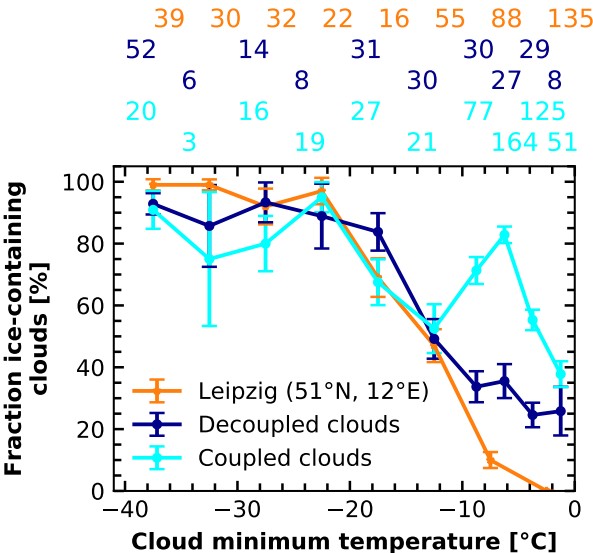

**Figure 5.** Same as Fig. 4 (a) with the Arctic clouds separated by their coupling state (cyan: coupled clouds, dark blue: decoupled clouds).

below 500 m, however, is also up to 70%. To further investigate if this effect may be linked to a possible INP source at the surface we separated the data set by the surface coupling state of the clouds, as described in Sect. 2. The resulting distribution for both surface-coupled (cyan) and -decoupled clouds (dark blue) is shown in Fig. 5. Between -15 and 0°C strong coupling effects can be seen. Surface-coupled ice-containing cloud intervals compared to decoupled ones occurred more frequent by a factor of 2-6 (e.g. 164 vs. 27 in number of observed clouds and 83% vs 36% in frequency of occurrence between -7.5 and -5°C minimum cloud temperature). Below -15°C this effect vanished and both curves show a similar distribution as found over Leipzig. Investigating lower minimum cloud temperature the number of cases of surface-coupled clouds reduces.

## 4  Discussion

This study found an influence of a thermodynamic linkage between the surface and the cloud and the heterogeneous ice formation within. As this study is one of the first to provide insights on the presented matter, a general discussion of the methodology and possible instrument effects on the ice detection and the determination of the coupling state is given here first.

### 4.1  Methodology and instrument effects

The presented analysis is based on well established methods developed and applied in previous studies such as Ansmann et al. (2009); Seifert et al. (2010, 2011, 2015); Kanitz et al. (2011) and the data was careful screened for ice occurrence solely using lidar data. Despite having a much more ice-sensitive Mira-35 cloud radar available, this was not utilized for ice detection. The reason behind this is the frequent occurrence of low-level clouds below the lowest radar range gate of 155 m above the

instrument. Such clouds were frequently observed during PS106 (25% of the observational time; Griesche et al., 2020) and hence cloud-radar-based statistics would be biased towards higher-level clouds. Nevertheless statistics based on the available cloud radar observations are shown in Appendix A. Quantitative differences are obvious compared to Fig. 5 but the effect of a larger fraction of ice containing clouds in the case of surface-coupling is still visible.

Oriented ice crystals can have a strong effect on the observed lidar signal. In the case of large, horizontally oriented plate crystals and a zenith-looking lidar the measured backscatter strongly increase while the depolarization ratio is close to zero (e.g. Noel and Sassen, 2005; Westbrook et al., 2010), the so-called specular reflection. Hence, most ground-based lidars are tilted off-zenith by a few degrees, in the case of Polly$^{XT}$ by 5°. This avoids most of the specular reflection effects, which otherwise may mask the presence of ice (Noel et al., 2002). Yet, Silber et al. (2018) showed that the effects are still visible up to an off-zenith angle of 10° in the case of planar ice crystals. Sassen and Takano (2000) found strong depolarization effects in case of the rare event of oriented columns for an off-zenith angle 1-2°. However, to date it has not been shown how randomly oriented columnar ice crystals, as they form above -10°C in the temperature regime where we found the strongest effect of surface coupling, influence the lidar signal at varying zenith angles. According to Noel et al. (2004), columnar crystals with high axis ratio can be expected to have high particle depolarization ratios on the order of 0.5.

The coupling approach has its limitations, as the height dimension is not considered and thus a larger fraction of lower clouds might be classified as surface-coupled. Therefore, the coupling retrieval was tested by setting a maximum potential temperature gradient between the surface and the liquid layer base for the surface-coupled state. Only these cloud profiles, where beside the original coupling requirement also the gradient between surface and liquid layer base was below a threshold of $2\,\mathrm{K km^{-1}}$, were considered as surface-coupled (see Appendix B). While using such a threshold the number of surface-decoupled cloud profiles increases at the expense of surface-coupled ones. Yet, the observed surface coupling effect on the ice occurrence remains. In Appendix B we also discuss a possible influence of the radiosonde measurement uncertainty on the surface coupling retrieval, which we have not considered otherwise in our analysis.

## 4.2 Possible causes for an increase of ice occurrence

The separation of the Arctic cloud data set revealed the presence of surface effects on the enhancement of the occurrence of ice formation. However, for clouds uncoupled from the surface, or at greater height and lower temperature, the ice frequency statistics are similar to what is observed over northern-hemisphere mid-latitude sites. This is consistent with previous findings that the free-tropospheric aerosol conditions in the Arctic are dominated by aged aerosol pollution mixed with dust and wildfire smoke from lower latitudes (e.g. Abbatt et al., 2019; Law et al., 2014; Willis et al., 2018, and references therein).

The reasons for the increase in ice forming efficiency for low and coupled clouds in the Arctic is most likely caused by effects resulting from the linkage to the surface (Solomon et al., 2014). A surface coupling of the cloud is accompanied with a well-mixed layer from the surface up to liquid layer base. Multiple processes were discussed in previous studies as potential candidates for explaining the observations. They will be listed and examined below.

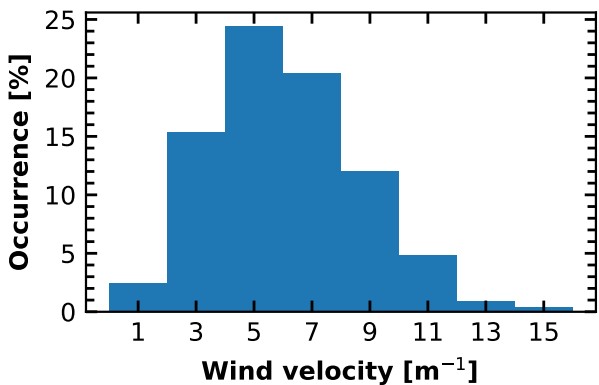

**Figure 6.** Histogram of the mean surface wind for any of the analyzed 30-min time interval.

In Shupe (2011) one reason for the high occurrence of low-level ice clouds was due to near-surface diamond dust. As this effect is stronger in winter and close to land (Intrieri and Shupe, 2004), it can be neglected as a dominating reason for the Arctic clouds during the investigated Polarstern cruise.

An enhanced fraction of ice-containing clouds could be attributed to blowing snow, as on the one hand, the lifted snow can be interpreted by the lidar as ice cloud. On the other hand blowing snow particles can act as seeds for ice crystals via the secondary ice multiplication processes (Rogers and Vali, 1987). Yet, an influence of blowing snow on the results can be ruled out. As Serreze and Barry (2014) pointed out, a minimum wind speed of $15\,\mathrm{ms}^{-1}$ is needed to lift the snow even a few meters above the ground. Since the wind speed during the PS106 campaign did not even reach this threshold (see Fig. 6), we are confident that blowing snow did not affect our findings. To consider also possible seeding effects from precipitating clouds above as found in Vassel et al. (2019), the data set has been filtered for those situations with little to no effects on the results.

Another potential explanation is the presence of an increased number of INPs in the surface-coupled clouds that are already active at -5°C. The aerosol distribution in the summertime Arctic marine boundary layer is influenced by local sources and downward mixing from the free troposphere of long-range transported aerosol (Willis et al., 2018). Typically only material of biogenic sources is active as INP at such high temperatures (Kanji et al., 2017). Biological aerosol from areas of open water within the marginal sea ice zone or from open leads or polynyas may be mixed into the coupled cloud layers where they can act as ice nuclei (Burrows et al., 2013) and increase the probability of ice production. The origin of such highly active INP needed for such an effect is still under discussion in the literature. Wex et al. (2019) found the largest INPC in the Arctic in summer with INP being active for temperatures up to -5°C. Hartmann et al. (2020) suggested these INP may be of biogenic origin from local marine sources such as open leads or polynyas. In an attempt to narrow down possible sources for marine INP in the Arctic, Ickes et al. (2020) compared the ice nucleation ability of Arctic sea surface micro layer samples and two different predominant Arctic phytoplankton species. Even though these samples showed ice nucleating activity already under moderate supercooling conditions, no clear evidence was found that they may serve as local marine INP source. Decoupling then again indicates a separation of the cloud from the surface due a stable layer below liquid base. Decoupled clouds, however, show

also slightly enhanced fraction of ice-containing clouds, e.g., compared to clouds observed above Leipzig. We do not know the life cycle of the cloud and its thermodynamic state prior to observation. Hence, any explanation for the reasons behind this observation would be speculative.

To examine potential enhancement of aerosol effects on the surface-coupled clouds, we performed a lidar-based aerosol analysis for PS106. Figure 7 shows profiles of the particle backscatter coefficient at 532 nm wavelength for 9 different time periods adjacent to cloud observations, when Polly$^{XT}$ was able to probe cloud free air masses. These profiles were derived using the procedures described by Baars et al. (2016). In situations of persistent cloud cover when no clear-sky lidar calibration was possible, calibration of the backscatter coefficient profiles was performed as described in Jimenez et al. (2020). The y-axis in Fig. 7 was normalized to the respective decoupling height of each individual profile. A reduction in the backscatter coefficient $\beta$ above the decoupling height can be seen. The sharp increase visible in some of the profiles is due to the liquid layer base, where the backscattered signal strongly increases. From the backscatter coefficient the lidar extinction coefficients can be estimated by using a lidar ratio of 50 sr for non-marine aerosol (Müller et al., 2007; Tesche et al., 2009; Groß et al., 2011). The extinction coefficient resulting from the profiles presented in Fig. 7 are much smaller compared to typical values for marine aerosol (40-100 Mm$^{-1}$; Kanitz et al., 2013; Bohlmann et al., 2018), except for the two profiles on 27 June 2017 (note that a typical lidar ratio for marine aerosol is <50 sr (Groß et al., 2011; Bohlmann et al., 2018) and would result in even smaller extinction coefficient values). Aeronet photometer measurements at Arctic sites show values for the Ångstrom exponent (440–870 nm) around 1.5 (not shown) indicating a fine-mode-dominated aerosol distribution. The Ångstrom exponent for typically coarse-mode-dominated marine aerosol centers around 0.5 (Smirnov et al., 2009; Yin et al., 2019). Both, the small values of extinction coefficient combined with the high Ångstrom exponents are clear indications that the observed aerosol is not of typical marine origin. Yet, an estimation of the INPC from the lidar measurements, as presented by Mamouri and Ansmann (2016) for different aerosol types is not possible, because the respective parameterizations for such biogenic aerosol in the Arctic is still missing.

For both coupling states at a minimum cloud temperature below -15°C the occurrence of ice-containing clouds increases strongly and reaches values close to 100% at temperatures below -25°C. A rather similar pattern was found over Leipzig (Kanitz et al., 2011). To provide further insights on this matter we performed an air mass source attribution analysis based on the Lagrangian particle dispersion model FLEXPART (Pisso et al., 2019), as introduced by Radenz et al. (2021). Throughout the analyzed period (1 June - 16 July 2017) particles were traced for 10 days, starting every 3 hours with height steps of 500 m. Based on the MODIS land cover classification from Broxton et al. (2014), 7 categories, namely 'water', 'forest', 'savanna/shrubland', 'grass/cropland', 'urban', 'snow' and 'barren' were defined as possible source regions (Radenz et al., 2021). The relative residence time of the particles below a reception height of 2 km above different possible aerosol source areas has been calculated and is shown in Fig. 8. A strong decrease of water as possible aerosol source region for particles arriving above 2 km is obvious, while snow/ice (note these are land based ice sheets and glaciers, not the frozen Arctic ocean), savanna/shrubland, grass/cropland and forest increase. This distribution agrees with what has been found for the continental site Krauthausen in Germany (Radenz et al., 2021, Fig. 15). This is confirmed by the common view that the aerosol in free troposphere of the Arctic is dominated by long-range transport.

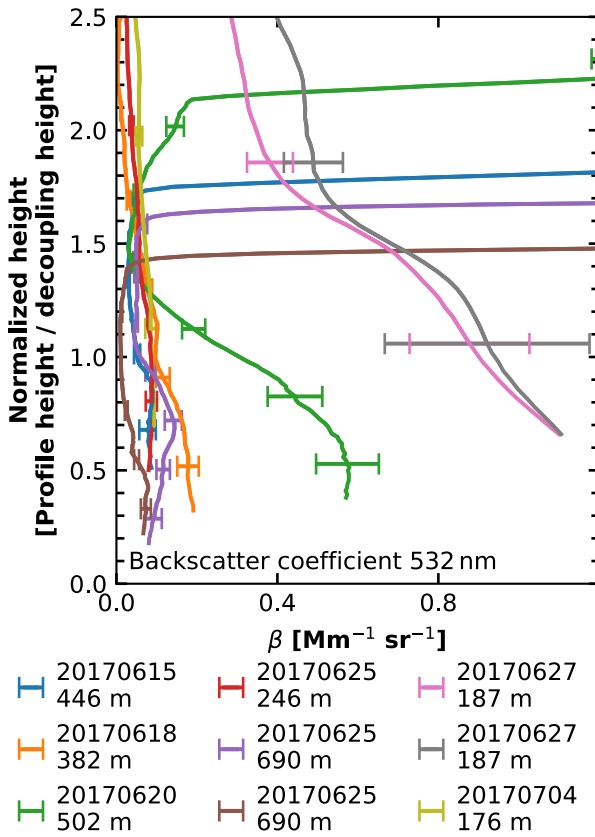

**Figure 7.** (a) Profiles of particle backscatter coefficient at 532 nm from Polly[XT]. The date of the corresponding profiles together with the respective decoupling height are annotated below.

After ruling out the possibility of blowing snow, seeding effects and diamond dust as reason for the observations, the only plausible explanation is highly active INPs which are more abundant in surface-coupled than in decoupled clouds. Given the mixture of the summertime marine layer aerosol in the Arctic and the high temperatures where the effect was observed, the most likely source for these INP is biogenic material from local marine sources.

The presented results are based on a two month campaign. And even though strong evidence was found that the effect is a consequence of aerosol of marine origin acting as INP in the cloud this rather short time coverage obviously can not cover all situations which are of interest in this matter. The influence of the distance to open water surfaces such as the open sea or leads and polynyas as well as a possible seasonal effect could not be studied.

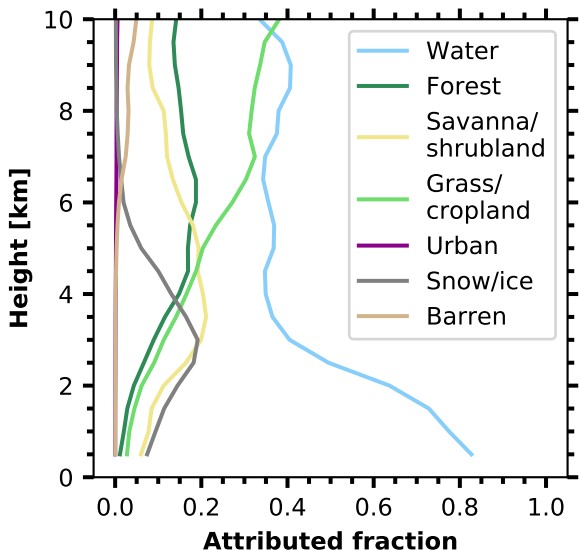

**Figure 8.** Fraction of residence time below the reception height (2 km) above different possible aerosol source regions based on a FLEXPART analysis.

## 5    Summary & Conclusions

In this study, differences in the fraction of ice-containing clouds for surface-coupled and decoupled clouds were investigated. In order to do so, lidar, cloud radar, and radiosonde observations from the RV Polarstern cruise PS106 around Svalbard in the Arctic summer 2017 were analyzed. Beside minimum cloud temperature, the data show a significant dependence of the liquid layer base height and coupling state of the cloud on the probability of ice formation. Figure 5 compares the fraction of ice-containing clouds for different minimum cloud temperatures for coupled and decoupled clouds. Strongest differences have been found at minimum cloud temperatures slightly below freezing. Above -15°C surface-coupled ice-containing clouds occur more frequently by a factor of 5 in numbers of observed clouds and by a factor of 2 in frequency of occurrence. Furthermore, the number of analyzed data is largest in this range, which underlines the significance of this finding. A similar ice cloud occurrence below -15°C as found over Leipzig together with a rather land-based dominated aerosol source attribution suggest the presence of continental aerosol in the free troposphere.

In the frame of our study, we examined the potential reasons for the surface coupling effects on cloud ice occurrence by means of a literature survey. However, seeding from higher ice clouds, as well as from blown snow or ice fog can be ruled out for the analyzed observation period. As a most likely explanation we found that the larger reservoir of marine ice nucleating particles in the surface-coupled marine boundary layer leads to higher freezing efficiency in the clouds which have at least their base in that layer. This conclusion is corroborated by recent in-situ based studies of the INPC which took place in close vicinity to open water surfaces in the marine Arctic boundary layer. Future studies hence should focus on the linkage between

types of aerosols raised to cloud level and the fraction of ice-containing clouds in order to confirm this hypothesis. It could also be worth investigating different Arctic and Antarctic cloud data sets with respect to their distance from the marginal ice zone, open leads and polynyas. If indeed INP from marine origin control heterogeneous ice formation that strongly, a decrease of this effect with increasing distance from open water should be detectable. For a better understanding of the phenomenon, measurements in different seasons and regions of the Earth should be made to determine if this effect is only characteristic for the Arctic summer.

*Data availability.* The lidar measurements are available by Griesche et al. (2019), the cloud radar measurements by Griesche et al. (2020). The radiosonde data is available by Schmithüsen (2017a) (PS106.1) and Schmithüsen (2017b) (PS106.2).

## Appendix A: Cloud radar for ice detection

To detect ice occurrence with the cloud radar, periods where the cloud radar showed enhanced reflectivity (as can be seen in Fig. 3 (b)) and linear depolarization ratio (LDR) were manually classified as ice-containing. The results are presented in Fig. A1. As expected due to the higher ice-sensitivity of the cloud radar, the detected ice cloud fraction overall increased. Yet, the observed effect remains the same: The surface-coupled clouds showed a stronger ice-cloud occurrence compared to the decoupled clouds. The effect is smaller than in Fig. 5. A reason for this could be that the lowest clouds, which are most likely effected by surface sources of INP, are not well represented in these statistics.

## Appendix B: Test of the coupling retrieval

The coupling retrieval was tested by setting a maximum potential temperature gradient between the surface and the liquid layer base for the surface-coupled state. This gradient was chosen as $\Delta\theta/\Delta z_{lim} = 2$ K/km, which was the lowest gradient we found in case of decoupled clouds using our approach to determine the surface coupling state. To test our approach only these clouds where in addition to the original requirement also the gradient between surface and liquid layer base was $\Delta\theta/\Delta z < \Delta\theta/\Delta z_{lim}$ were considered as surface-coupled. In this case the number of surface-decoupled cloud intervals increased at the expense of the surface-coupled ones (Fig. B1). Also the fraction of ice-containing surface-decoupled clouds increased, although, not as much as the fraction of ice-containing surface-coupled clouds. Thus, the surface-coupling effect is still obvious and even extends towards lower temperatures. However, the number of surface-coupled cloud intervals below -15°C is rather small and the findings here should be considered with care.

The measurement uncertainty in temperature and pressure of the radiosonde RS-92 is given by Vaisala to be $\Delta$T=0.5°C and $\Delta$p=1hPa (Jensen et al., 2016). As we compare observations within one profile, systematic errors can be neglected and only the standard deviation of differences in pairs of soundings is of interest. This was given by Jensen et al. (2016) as $\sigma_T = 0.2$°C (<100hPa) and $\sigma_p = 0.3$ hPa (<100hPa). Using error propagation we estimate the error in the potential temperature $\Delta\theta$. This error was used to calculate the surface coupling state by varying the threshold for the coupling state (0.5 K $\pm \Delta\theta$). The results

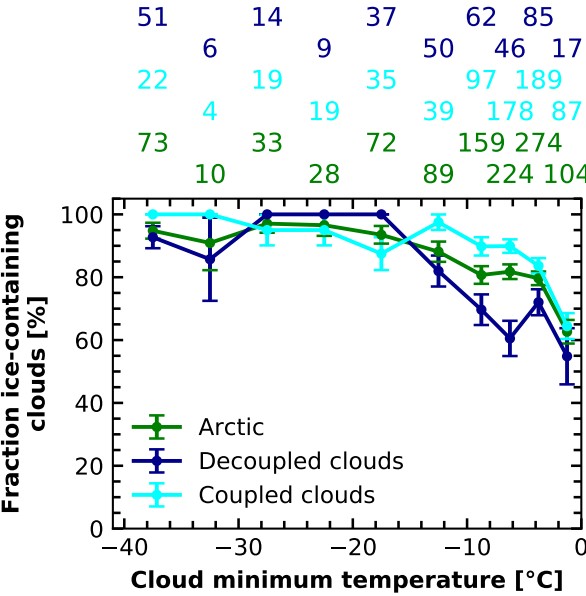

**Figure A1.** Same as Fig. 4 but using the cloud radar for ice detection. In green the results of the complete data set is shown, in cyan for the coupled clouds and in dark blue for the decoupled clouds.

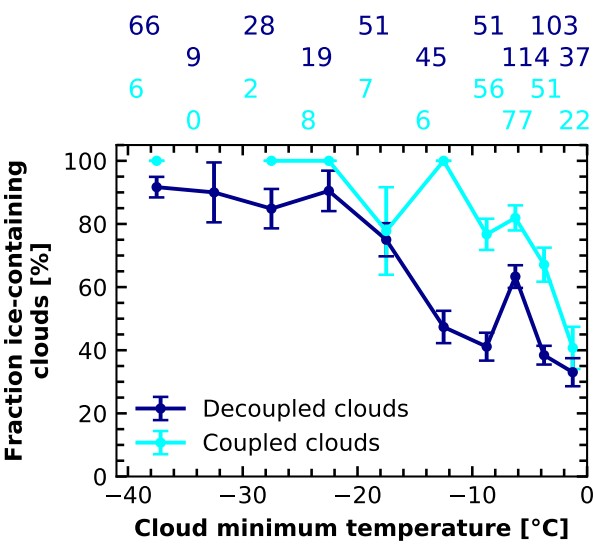

**Figure B1.** Same as Fig. 4 but using also a threshold in the potential temperature gradient to identify surface-coupling.

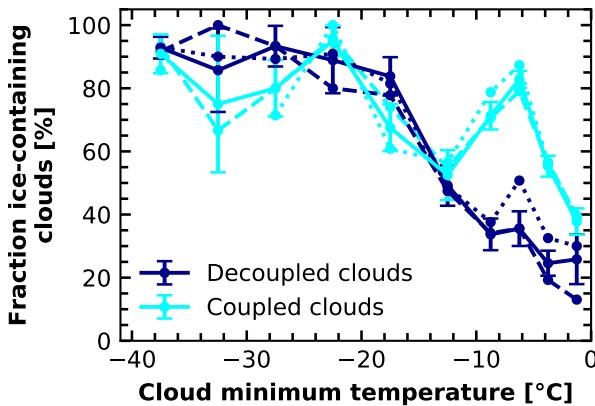

**Figure B2.** Continuous lines same as Fig. 4. The dashed lines represent the respective results for the upper end of the error margin (i.e. profiles were classified as decoupled using a threshold for the coupling state of 0.5 plus the error in $\theta$) and the dotted lines represent the lower end of the error margin (i.e. profiles were classified as decoupled using a threshold for the coupling state of 0.5 K minus the error in $\theta$).

are shown in Fig. B2. The variation of the threshold resulted in a change of the number of coupled cloud intervals. In the case of 0.5 K + $\Delta\theta$, 31% of the originally surface-coupled analyzed cloud intervals were classified as decoupled, while for 0.5 K - 410 $\Delta\theta$, 7.5% of the originally surface-decoupled intervals were classified as coupled. Yet, the observed effect on the fraction of ice-containing clouds was within the statistical uncertainty of the original approach and thus the measurement uncertainty of the radiosondes was not further considered in our analysis.

*Author contributions.* HG and KO conducted the ice-containing cloud analysis under supervision of PS. HG performed the lidar analysis under supervision of AA, PS and RE. MR performed the back-trajectory analysis. HG, KO and PS prepared the manuscript. HG finalized the 415 manuscript under supervision of AA and PS.

*Competing interests.* The authors declare that they have no conflict of interest.

*Acknowledgements.* We gratefully acknowledge the funding by the Deutsche Forschungsgemeinschaft (DFG, German Research Foundation) – Project Number 268020496 – TRR 172, within the Transregional Collaborative Research Center "ArctiC Amplification: Climate Relevant Atmospheric and SurfaCe Processes, and Feedback Mechanisms (AC)3". We thank the Alfred Wegener Institute and R/V Polarstern crew 420 and captain for their support (AWI_PS106_00). Authors acknowledge also support through ACTRIS-2 under grant agreement no. 654109 from the European Union's Horizon 2020 research and innovation programme.

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
