# Peer review of "Contrasting ice formation in Arctic clouds: surface coupled vs decoupled clouds"

_Atmospheric Chemistry and Physics, 2020_

## Referee Comment (RC1) · Anonymous Referee #1 · 4 Dec 2020

This study examines the impact of the surface coupling state of polar clouds observed during the PS106 voyage on the probability of ice occurrence. The authors use a comprehensive set of ground-based measurements to detect cloud boundaries, detect the presence of ice, and analyze the thermodynamic state of the atmosphere. The authors suggest that cloud coupled to the surface contain ice more frequently than decoupled clouds because of enhanced INP concentrations transported from below.

The manuscript provides a comprehensive introduction with a reasonable literature survey. A common change of tense in the introduction and other parts of the manuscript (e.g., p. 4 l. 3-30, p. l. 7-10) could impact to flow of the text to some readers, and hence, I recommend the authors to revisit this issue.

While I agree with the general conclusions made by the authors, I find multiple weak-

nesses in the methodology, which could significantly impact the analysis quantitative results, even if the conclusions will be similar. Therefore, I recommend major revisions of this manuscript before it should be considered for publication in ACP.

Major comments:

- The authors have the 35 GHz ground-based radar data at their disposal, one of the best remote-sensing instruments for the detection of precipitating hydrometeors (and ice particles in particular), even in very small concentrations. Yet, they only use the lidar data to detect precipitation. Commonly occurring cases of weak Arctic precipitation can be missed by lidars in such cases, as a result of the potentially minor contribution of very small ice concentrations to an air volume's total cross-sectional area of scatterers (e.g., when including nearly spherically-shaped aerosols). This is also evident in Fig. 2, where there is a clear indication of (weak) precipitating fall streaks in the radar data between 18-21 UTC (also suggested by the depolarization plot), even though this period is classified as an ice-free cloudy period. I think that the authors should incorporate the radar data in their analysis, because at the moment, weakly precipitating clouds could significantly change their analysis results (see for example Fig. 3 in Buhl et al., 2013; https://agupubs.onlinelibrary.wiley.com/doi/full/10.1002/grl.50792) in the proceeding figures.

- In continuation of the previous comment regarding ice detection using lidar depolarization ratio data, the analysis could have been influenced by specular reflection from plate ice crystals within the -20 – (-8) °C temperature range. In these cases where plates precipitate from the cloud base, the determined cloud base might be lower than it actually is (depending on the depolarization threshold), and a cloud can be classified as ice-free since the change in depolarization or the depolarization threshold for ice detection (not clear from the text) is not strong enough. I know that the common tilting of lidars by a few to several degrees off zenith (e.g., 5 degrees as in the PS106 voyage) is commonly believed to address this specular reflection issue, but that is a common misconception, as it does not consider commonly observed

higher canting angles of ice particles (see for example Appendix A in Silber et al., 2018; https://agupubs.onlinelibrary.wiley.com/doi/full/10.1029/2017JD027840, Noel et al., 2002; https://agupubs.onlinelibrary.wiley.com/doi/abs/10.1029/2002GL014828).

- I find the theta criterion for the determination of surface coupling state problematic because it doesn't consider the cloud height above the surface, which could result in more lower-level clouds being classified as coupled. As an example, a cloud base at 200 m with theta difference just below 0.5 K (e.g., theta of 260 K at the surface rising linearly to roughly 261 K at cloud base) would be considered coupled even though dtheta/dz = 5 K/km, which is strongly stratified. In their analysis, the authors should take into consideration the height dimension as well as the measurement uncertainty of the RS-41 radiosondes (0.3 C in T, 4% in RH). The current potential for a low-level coupled cloud bias could contribute to the stark coupled vs. decoupled ice occurrence fraction differences at higher temperatures discussed in Fig. 4, given the summertime dataset manifested in the greater occurrence of lower, warmer clouds (see for example the results from Svalbard in Nomokonova et al., 2019; https://acp.copernicus.org/articles/19/4105/2019/).

- Estimation of the INP number concentration: the method in Mamouri and Ansmann (2016) relied on European and Mediterranean data of aerosol mixtures, the values of which can be significantly different from Arctic regions (see for example Kanji et al., 2017, https://journals.ametsoc.org/mono/article/doi/10.1175/AMSMONOGRAPHS-D-16-0006.1/28236). Moreover, in the Arctic alone it has been shown that there is high INP variability and that INP concentrations are not correlated with multiple types of aerosols (e.g., Wex et al., 2019, https://acp.copernicus.org/articles/19/5293/2019/). Given the fact that based on our current knowledge INP occupy only a small fraction of the total aerosol number concentrations (and likely their projected area), there are just too many degrees of freedom in the INPNC retrieval and I do not see how can the authors estimate the INPNC even with the scaling factor they decided to use, and do not see how their conclusions could be dismissed even without the rather short INPNC

analysis discussion. I find it very hard to believe that the uncertainties in INPNC (as shown in Figure 6) are smaller than an order of magnitude.

Minore comments:

- There is no information on the route of the PS106 voyage. I recommend adding a map for reference or at the very least specify the latitude/longitude ranges of that voyage.

- p. 1 l. 2 - suggest defining that OCEANET is a platform in the first instance.

- p. 1 l. 11-12 "This provides further evidence ..." – this sentence is not supported by the analysis and is not explicitly discussed in the text. I recommend removing it or revising the analysis and text accordingly.

- p. 1 l. 13 – "acting as seeds for ice multiplication" – again, this impact of seeding from below is not discussed and supported by the text (might be suggested only implicitly in the discussion about blowing snow).

- p.1 l. 16-18 - suggest reordering these two sentences.

- p. 1 l. 23 - "above the one" - suggest rewording

- p. 3 l. 34 - "as it is the case for the 35-GHz ..." - Even though the ARM KAZR is a valid example, I recommend either removing this part of the sentence or providing a different example, because the KAZR nor the Barrow site are not discussed in this paper.

- p.3 l.21 - "First studies ..." - I do not understand this sentence - suggest rewording

- p. 3 l.27-29 - is this *the main feature* of Arctic clouds, or simply one of their common features? Also, do the clouds necessarily form in inversions, or do they form the inversions? I think that both options are plausible (see for example Morrison et al., 2012; https://www.nature.com/articles/ngeo1332, Silber et al., 2020; https://agupubs.onlinelibrary.wiley.com/doi/full/10.1029/2020GL087099, Sedlar, 2014; https://journals.ametsoc.org/view/journals/apme/53/12/jamc-d-14-

0065.1.xml?tab_body=fulltext-display)

- p. 6 l.3 - please clarify whether the linear or circular depolarization ratios are used (I suspect the former).

- p. 9 l. 31 - it could be the vast majority of clouds (> 80%) but this is certainly not every cloud.

- What are the depolarization thresholds for the determination of liquid and ice? Are there backscatter thresholds as well? These values should be explicitly specified for reproducibility by potential readers.

- p. 7 l. 1 - what is the slope or the metric with which cloud top is defined? This should also be specified.

- p. 7 l. 5 - "coldest temperature" - temperatures can lower but not colder - suggest rewording here and in other locations in the text.

- p. 7 l. 6-9 - This method of using the inversion base temperature as cloud top temperature may explain some of this study's results, as the assumption becomes less valid in cases where clouds protrude into temperature inversions, which often occur concurrently with stronger mixing, not necessarily down to the surface. I think that in the context of this paper the authors might be able to make their point by defining their current "cloud top temperature" as "minimum cloud temperature", which would also be valid for cloud protruding into an inversion, and would retain the essence of INP activation temperature widely discussed in the text. Also, note that note all polar liquid-bearing clouds are capped by a temperature inversion. See for example Sedlar and Tjernström, 2009; https://link.springer.com/article/10.1007/s10546-009-9407-1, Sedlar et al., 2012; https://journals.ametsoc.org/view/journals/clim/25/7/jcli-d-11-00186.1.xml?tab_body=fulltext-display, Sotiropoulou et al., 2014; https://acp.copernicus.org/articles/14/12573/2014/, Silber et al., 2020; https://agupubs.onlinelibrary.wiley.com/doi/full/10.1029/2020GL087099

[Figure]

- p. 7 l. 12-14 - based on the fact that the authors have used the radar for this seeding cloud proximity criterion, I think that they refer here to overlying hydrometeors rather than overlying clouds. If that is correct I recommend revising the text accordingly.

- p. 9 l. 31 - it could be the vast majority of clouds (> 80%) contain ice but this is certainly not every cloud as currently stated in the text.

- Fig. 3 caption confusion - deltaT should be 5 C below -10 C and vice versa.

- p. 11 l. 13-14 - "The reasons for the increase in ice forming efficiency for low and coupled clouds in the Arctic must be caused..." - while this is likely the case, I think that the authors should tone down this sentence.

- p. 12 l.10-11 - decoupling does not necessarily mean that there is an underlying inversion, but only that the underlying layer is stable. I suggest revising the text accordingly.

- p. 12 l. 11-13 - clouds largely act to destabilize the polar atmosphere and not the opposite. Another more likely possibility is that once the marine aerosols are mixed aloft, the atmosphere becomes decoupled as a result of radiative cooling of the surrounding ice surfaces.

- p. 12 l. 16 - add "as" before "such"

- p. 12 l. 23 - define beta

- p. 12 l.13 - a reduction in beta is generally seen throughout the atmospheric profile regardless of the decoupling height (and sometimes increases above the decoupling height such as in the green and blue curves), so I find this argument by the authors to be rather subjective.

- Fig. 6 - what do the normalized 0 and 2 values represent?

- p. 13 l.1 - Temperature units are missing.

---

## Referee Comment (RC2) · Anonymous Referee #2 · 16 Dec 2020

The manuscript by Griesche et al. describes an approach based on the synergy among the measurements provided by different ground based remote sensing sensors techniques (lidar, radar, radiosondes mainly) to study the ice content of surface-coupled and decoupled clouds in the Arctic using the data from a cruise of the Polarstern vessel in the Summer of 2017.

The authors objective is demonstrate or at least provided concrete support to the hypothesis that INPs from marine biological reservoir controls the ice heterogeneous nucleation of the Arctic clouds. This hypothesis is based on previous studies available in literature. More specifically, the authors aim at demonstrating that the dynamical and microphysical structure of surface-coupled cloud differs from the decoupled clouds with the first type being dominant in terms of frequency of occurrence and because more

frequently contain ice than the decoupled clouds (by a factor of 5). This hypothesis appears to be demonstrated from the results reported in the manuscript for the considered dataset. The dataset is limited one month and half in 2017: despite the fact it must be acknowledged the considerable effort spent to collect the presented measurements, a dataset with a longer time coverage covering at least two seasons - discussed also in conjunction with a more detailed meteorological analysis - could provide more robust results. The effect of the limited dataset time coverage may have an effect on the discussed results and this should be considered by the authors while it is faintly mentioned at the end of the paper only.

Below, I provide my general and a few specific comments. I recommend a major revision of the manuscript.

The presented approach lack of many details and the applied criteria often rely on assumptions inferred from the literature: these appear a bit forced or not supported by evidence or sensitivity studies which can quantify the related uncertainties. I list here the number of missing details which do not allow to quantify the uncertainty range in the statistical analysis and do not ensure the full reproducibility of the presented approach.

In the identification of ice clouds (section 2.1 Ice-containing cloud analysis), the description of the procedure applied to classify and characterize individual cloud profiles is purely qualitative, the thresholds applied to the value of the depolarization and backscattering coefficient are not mentioned indicating that the profiles have been evaluated on a subjective basis. There is no mention to the uncertainties and assumptions done in the lidar data processing (use of lidar ratios, calibration of profiles, , quantification of effects like specular reflection, etc. . .) which are quite relevant for the presented statistics. Everything could be referred to a literature paper to clarify the data processing, but, as it stands, I am not able to find one reference for these aspect in the entire section, only one for the multiple scattering affecting the depolarization ratio. The authors uses "the cold side of the temperature inversion which is closest to the cloud-radar-derived cloud top height in the radiosonde data to defined the cloud-top

temperature." It is not clear to me how large is the difference in meter between cloud top derived from the radar and the height of the radiosonde in correspondence of the cloud-top temperature. May a large difference be the result of a collocation effect which is negligible or not? In section 2.2, a scaling factor for the parameterization of DeMott et al. (2015) is derived from a single paper in literature, Gong et al. (2020), where filter samples from the Cape Verde Atmospheric Observatory were studied and INP active at temperatures above -10°C were found, which consists likely of biological material. This factor is assumed as a sort of "true" to estimate the INP concentration without any study on the sensitivity of the results to this assumption. The considered assumption may lead to large uncertainties in the retrieved INP profiles. The authors should not forget that the lidar retrieval have already uncertainties and is based on assumptions the effect of which might be amplified by this further assumption in the parameterization of mineral dust.

To conclude this point, I think Section 2.1 must be substantively reshaped.

2.From the text, It seems that the authors did not try a quantification of the effect on their approach of the presence of different types of aerosol beneath the clouds. No information are provided about the aerosol types (from data itself or transport model data), assuming the measurement platform necessarily implies the presence of biological aerosol mainly. Previous studies available in literature showed that the types of aerosol observed in the Arctic may be of very different origin depending on the air mass advection from lower latitudes and on the related natural or antropogenic events (dust, biomass, . . .). Volcanic ares with intense activities throughout the year must be also considered as an important potential source. Why did the authors not used at least the lidar data to type the observed aerosols (for example using lidar color ratio and depolarization) or transport models?

3. Likewise It's unclear why the authors did not use the cloud radar measurements in the identification and filtering of cases with ice crystal precipitation. This is another points which can change the statistics collected in too subjective way, to my opinion,

affecting the final results

4. For the results shown in Figure 4, the reported statistics on the number of profiles considered in the statistics poses a questions on the dependence of the results from dataset time coverage: is the number of coupled ice cloud profiles much higher because these are the most recurrent cases for the investigated period of the year? This aspect must be discussed in clear way, maybe using ancillary datasets.

I think also Section 4 must be improved.

Specific comments Line 6 page 1: replace "in " with "within"

Line 9 page 1: the factor mentioned here in in the range 2-5, but it is not mentioned in which temperature range assumes these values. It becomes clearer from the following sentence. Please rephrase.

Lines 14-16 page 1: this sentence is not appropriate for the abstract but for the discussion section, please remove.

Line 8 page 6: "to date" must be at the end of the sentence.

Line 8 page 2: "yet" date must be at the end of the sentence.

Line 24 page 3: remove higher at the beginning of the line and change ". . . than do. . ." with "higher than".

Page 7: Figure caption please put "yet" at the end of the sentence of replace "an" with "a".

Page 7 line 1: it is not clear to me which algorithm has been used to retrieve the cloud top height from the radar measurements, please specify.

Page 7 line 13: the detection of cloud by the radar up to the tropopause maybe depending on the size of ice crystals and by the concurrent atmospheric attenuation. Please nuance this sentence.

Page 7 line 17: please replace "simplified coupling algorithm" with "simplified version of the algorithm".

Page 10 line 8, page 11 line 1-2: do you have any reference to support your arguments?

Page 12 line 1: "is the strongest"or "is stronger"?

Page 12 line 11-13: in this part of the manuscript, there is often the comparison with statistics collected in Leipzig; I am wondering if the authors can say a few words on the usage of data from one site only at the mid-latitudes to make the comparison with a more stable region like the Arctic.

Page 14 lines 14-19: in this paragraph, the reader can find the list of the limitation of the results presented in this study. These are highlighted only at the end of the manuscript as an outlook for future studies while they should be discuss also when the results are presented.
* * *

---

## Author Comment (AC1) · 15 Mar 2021

**Author's response to two anonymous reviews for ACP-2020-1096**

We thank the two referees for their time and providing us with their comments and ideas, which improved the quality of the manuscript. We have revised the manuscript considered the points raised by the reviewers and provide here a point-to-point answer to the single remarks (the referee comments are highlighted in blue-italic). The page, line and figure numbering refer to the revised manuscript. Additionally a diff-version of the manuscript is provided for tracking the changes.

**Specific reply to Referee #1**

**Major comments:**

*- The authors have the 35 GHz ground-based radar data at their disposal, one of the best remote-sensing instruments for the detection of precipitating hydrometeors (and ice particles in particular), even in very small concentrations. Yet, they only use the lidar data to detect precipitation. Commonly occurring cases of weak Arctic precipitation can be missed by lidars in such cases, as a result of the potentially minor contribution of very small ice concentrations to an air volume's total cross-sectional area of scatterers (e.g., when including nearly spherically-shaped aerosols). This is also evident in Fig. 2, where there is a clear indication of (weak) precipitating fall streaks in the radar data between 18-21 UTC (also suggested by the depolarization plot), even though this period is classified as an ice-free cloudy period. I think that the authors should incorporate the radar data in their analysis, because at the moment, weakly precipitating clouds could significantly change their analysis results (see for example Fig. 3 in Buhl et al., 2013; https://agupubs.onlinelibrary.wiley.com/doi/full/10.1002/grl.50792) in the proceeding figures.*

Indeed the cloud radar Mira-35 is much more sensitive to ice detection as the lidar. As mentioned in Bühl et al. (2013) the lidar has a detection threshold in IWC of about $10^{-6}$ kg m$^{-3}$. But, as shown in Griesche et al. (2020) frequently low clouds were observed during the analyzed campaign with a cloud base below the lowest detection range of the radar (155m above the instrument). In some cases, even the cloud top was below this height. These clouds would have been falsely classified using the cloud radar only, while these clouds are actually those closest to the surface and therefore most likely coupled to it. Using the near-field capabilities of our polarization lidar Polly$^{XT}$ is thus a prerequisite for the presented coupling study. Another point is that we wanted to do a study that is comparable to previous studies, as the one of Kanitz et al. (2011). Nevertheless, we created the same analysis of the ice-containing clouds for surface-coupled and –decoupled cases (see Fig. 1). As to be expected the amount of ice containing clouds increased, yet ice-containing surface-coupled profiles were both absolutely (i.e. the numbers of profiles) and relatively (i.e. the fraction of ice-containing clouds) more frequent. To conclude this point: we think that the challenges we would be facing including the cloud radar for the ice detection would have deformed the statistics by excluding the very near clouds, and would have made a comparison to other studies difficult. And last but not least, the observed effect (more surface-coupled ice-containing cloud profiles with a IWC down to $10^{-6}$ kg m$^{-3}$) would not be affected by radar observations. To discuss this and the following points regarding the methodology, we added the Subsection *4.2 Methodology and instrument effects* in the Discussion Section (the usage of the lidar for the ice detection is discussed on page 15, line 1-6). Additionally the results are presented in Appendix A.

[Figure]

**Figure 1:** Fraction of ice-containing clouds determined using the cloud radar for ice detection. In green the results of the complete data set is shown, in cyan for the coupled clouds and in dark blue for the decoupled clouds.

*- In continuation of the previous comment regarding ice detection using lidar depolarization ratio data, the analysis could have been influenced by specular reflection from plate ice crystals within the -20 – (-8) °C temperature range. In these cases where plates precipitate from the cloud base, the determined cloud base might be lower than it actually is (depending on the depolarization threshold), and a cloud can be classified as ice-free since the change in depolarization or the depolarization threshold for ice detection (not clear from the text) is not strong enough. I know that the common tilting of lidars by a few to several degrees off zenith (e.g., 5 degrees as in the PS106 voyage) is commonly believed to address this specular reflection issue, but that is a common misconception, as it does not consider commonly observed higher canting angles of ice particles (see for example Appendix A in Silber et al., 2018; https://agupubs.onlinelibrary.wiley.com/doi/full/10.1029/2017JD027840, Noel et al., 2002; https://agupubs.onlinelibrary.wiley.com/doi/abs/10.1029/2002GL014828).*

We agree that the issue of specular reflection is not discussed well enough in the manuscript. However, since this effect is most prominent for rather large dendrites, which form at a temperature of roughly -15°C, we think specular reflection has little influence on our findings where we found the strongest influence at a temperature above -10°C. We extended the Discussion section and elaborated the effect in more detail (see page 15, line 7 and the following).

*- I find the theta criterion for the determination of surface coupling state problematic because it doesn't consider the cloud height above the surface, which could result in more lower-level clouds being classified as coupled. As an example, a cloud base at 200 m with theta difference just below 0.5 K (e.g., theta of 260 K at the surface rising linearly to roughly 261 K at cloud base) would be considered coupled even though dtheta/dz = 5 K/km, which is strongly stratified. In their analysis, the authors should take into consideration the height dimension as well as the measurement uncertainty of the RS-41 radiosondes (0.3 C in T, 4% in RH). The current potential for a low-level coupled cloud bias could contribute to the stark coupled vs. decoupled ice occurrence fraction differences at higher temperatures discussed in Fig. 4, given the summertime dataset manifested in the greater occurrence of lower, warmer clouds (see for example the results from Svalbard in Nomokonova et al., 2019; https://acp.copernicus.org/articles/19/4105/2019/).*

To test the influence of the theta gradient on the coupling retrieval, we set a gradient-threshold for the coupled state. The threshold was set to the minimum gradient between surface and liquid layer base observed during this study in the case of decoupling, which was dtheta/dz = 2 K/km. In Figure 2, the retrieved results are shown. As expected the number of decoupled profiles increased at the

expense of coupled profiles. And with the number of profiles also the frequency of occurrence of surface-decoupled ice-containing profiles increased. Yet, in case of surface coupling the frequency of occurrence of ice clouds increases even stronger. Hence, those profiles, which were classified as coupled with our original retrieval but decoupled when the gradient-threshold is considered, have a higher fraction of liquid-only than ice-containing clouds (increase in frequency of occurrence of surface-coupled ice-containing profiles). Still, the fraction of ice-containing clouds within the reclassified profiles is higher, compared to the originally surface-decoupled classified profiles (hence, also an increase in frequency of occurrence of surface-decoupled ice-containing profiles). In addition (not shown but as predicted by the reviewer) the reclassification only concerns clouds with a liquid cloud base at or below 400m (with the majority at 200m or below). The proximity of those clouds to the surface increases the likelihood of an effect of surface originated aerosols. We conclude that, despite the deficiencies of our original approach to determine the surface coupling state, we stick to this retrieval as this was already used in Gierens et al. (2020). We have discussed these points in the manuscript on page 16, line 6-11 and presented the results in Appendix B.

[Figure]

**Figure 2:** Fraction of ice-containing clouds determined using also a threshold in the potential temperature gradient to identify surface-coupling (cyan surface-coupled clouds and dark blue decoupled clouds).

The measurement uncertainty of the radiosonde can be split into systematic and random errors. Comparing values within one radiosonde profile, a systematic uncertainty is rather negligible, as it is correlated throughout the profile. Random uncertainties on the other hand, can vary in sign and magnitude for each point. The standard deviation between twin soundings up to 100hPa are given by Vaisala for the RS-92 (which was still used during PS106) for zemperature as 0.2°C and for pressure as 0.5 hPa hPa (Jensen et al., 2016). To test how this might influence the coupling retrieval, we used error propagation to determine the uncertainty in the potential temperature profile $\Delta\theta$. The error was used to estimate an upper and lower threshold boundary of the coupling retrieval (0.5K ±$\Delta\theta$) and to repeat the coupling retrieval with these two values. The effect on the fraction of ice-containing clouds for the coupled and decoupled cases was located within the statistical uncertainty of the results (Fig. 3). Hence, we stick to our original analysis, but made the reader of the manuscript aware that the measurement uncertainty of the radiosonde has not been considered (see page 16, line 11-13 and Appendix B)

[Figure]

**Figure 3:** Continuous lines same as Fig. 4 of the manuscript. The dashed lines represent the respective results for the upper end of the error margin (i.e.profiles were classified as decoupled using a threshold for the coupling state of 0.5K +$\Delta\theta$) and the dotted lines represent the lower end of the error margin (i.e. profiles were classified as decoupled using a threshold for the coupling state of 0.5K -$\Delta\theta$).

*- Estimation of the INP number concentration: the method in Mamouri and Ansmann (2016) relied on European and Mediterranean data of aerosol mixtures, the values of which can be significantly different from Arctic regions (see for example Kanji et al., 2017, https://journals.ametsoc.org/mono/article/doi/10.1175/AMSMONOGRAPHS-D-16-0006.1/28236).*
*Moreover, in the Arctic alone it has been shown that there is high INP variability and that INP concentrations are not correlated with multiple types of aerosols (e.g., Wex et al., 2019, https://acp.copernicus.org/articles/19/5293/2019/). Given the fact that based on our current knowledge INP occupy only a small fraction of the total aerosol number concentrations (and likely their projected area), there are just too many degrees of freedom in the INPNC retrieval and I do not see how can the authors estimate the INPNC even with the scaling factor they decided to use, and do not see how their conclusions could be dismissed even without the rather short INPNC analysis discussion. I find it very hard to believe that the uncertainties in INPNC (as shown in Figure 6) are smaller than an order of magnitude.*

We have decided to remove the Section of the INPC estimation together with Figure of the INPC (Fig. 6b in the old manuscript). The reason for this is the great uncertainty of the presented approach. Instead, we extended the discussion on the measured attenuated backscatter coefficient values, including the lack of parametrizations for INP in the Arctic (see page 14, line 1-16).

**Minor comments:**

*- There is no information on the route of the PS106 voyage. I recommend adding a map for reference or at the very least specify the latitude/longitude ranges of that voyage.*
A Figure with the PS106 track and dates is added.

*- p. 1 l. 2 - suggest defining that OCEANET is a platform in the first instance.*
Done.

*- p. 1 l. 11-12 "This provides further evidence : : :" – this sentence is not supported by the analysis and is not explicitly discussed in the text. I recommend removing it or revising the analysis and text accordingly.*
We complemented the discussion by a back-trajectory analysis (page 14, line 17-25).

*- p. 1 l. 13 – "acting as seeds for ice multiplication" – again, this impact of seeding from below is not discussed and supported by the text (might be suggested only implicitly in the discussion about blowing snow).*

We considered this point more explicitly in the discussion.

*- p.1 l. 16-18 - suggest reordering these two sentences.*
The Section about the INPC estimation has been removed from the manuscript.

*- p. 1 l. 23 - "above the one" - suggest rewording*
Done.

*- p. 3 l. 34 - "as it is the case for the 35-GHz ..." - Even though the ARM KAZR is a valid example, I recommend either removing this part of the sentence or providing a different example, because the KAZR nor the Barrow site are not discussed in this paper.*
We have raised this point using the ARM KAZR because the limitation to detect the most-likely surface-coupled clouds below 150 m height concerns also this instrument. We consider this point as important because it hinders one to use the temporally extensive ARM datasets to be utilized for coupling studies similar to the one presented in here.

*- p.3 l.21 - "First studies ..." - I do not understand this sentence - suggest rewording*
The paragraph has been reframed.

*- p. 3 l.27-29 - is this \*the main feature\* of Arctic clouds, or simply one of their common features? Also, do the clouds necessarily form in inversions, or do they form the inversions? I think that both options are plausible (see for example Morrison et al., 2012; https://www.nature.com/articles/ngeo1332, Silber et al., 2020; https://agupubs.onlinelibrary.wiley.com/doi/full/10.1029/2020GL087099, Sedlar, 2014; https://journals.ametsoc.org/view/journals/apme/53/12/jamc-d-14-065.1.xml?tab_body=fulltext-display)*
We reworded the paragraph and also considered nonturbulent formation of Arctic clouds.

*- p. 6 l.3 - please clarify whether the linear or circular depolarization ratios are used (I suspect the former).*
The volume depolarization was used, which is derived from the linear depolarization ratio measurement of the lidar.

*- p. 9 l. 31 - it could be the vast majority of clouds (> 80%) but this is certainly not every cloud.*
We reworded that sentence stating that the majority of clouds contains ice.

*- What are the depolarization thresholds for the determination of liquid and ice? Are there backscatter thresholds as well? These values should be explicitly specified for reproducibility by potential readers.*
Similar to a comparable concern raised by reviewer 2 we point to the difficulty in separating ice and liquid clouds by the lidar volume depolarization. There have not been many studies on a quantification of the lidar volume depolarization on ice detection, as volume depolarization ratio is the superposition of molecular and particulate backscatter in the co- and cross-channels. To tackle this obstacle we used manpower to manually analyze the data set and provided a detailed explanation of the methodology.

Therefore we decided to describe the applied method in detail. We analyzed the complete available data set. The only periods that have been excluded from the analysis, are as described those with favorable seeding conditions (i.e., a cloud above the analyzed cloud within 1km). Combined with the fact that all data is freely available, this study should be reproducible by anybody given the description of the methodology in the article.

Nevertheless we made a first attempt to provide a depolarization threshold on the ice detection, which is shown below. We calculated a minimum volume depolarization where the lidar should be able to detect ice. This is based on an ice water content detection threshold of $10^{-6}$ kg m$^{-3}$ (Bühl et al., 2013) which was converted into lidar extinction using the approach of Hogan et al. (2006). Using a lidar ratio of 30 sr (typical single-scattering lidar ratio of ice crystals, see, e.g., Seifert et al., 2007) we calculated

the particle backscatter coefficient. The molecular backscatter coefficient at 532-nm (wavelength of the used depolarization channels) was derived using scattering theory (Hinkley, 1976) for a temperature of -10°C and air pressure of 925 hPa (ca. 700 m height). Assuming a particle depolarization ratio of ice crystals of 0.5, a minimum volume depolarization of 0.03 was found corresponding to the ice detection threshold of $10^{-6}$ kg m$^{-3}$.

This threshold was used to reproduce our study with an automatic approach. We defined a volume depolarization signal above 0.03 in four contiguous lidar range gates, but below the liquid-cloud base, as an ice layer. An ice-containing cloud profile was defined, when during half of the profile time an ice layer with volume depolarization ratio > 0.03 was detected. The results are presented in Fig. 4. While we found minor quantitative differences between the manually analyzed data set below -15°C, the main message of this manuscript remains: The occurrence of ice-containing surface-coupled cloud profiles at temperatures above -10°C is much higher compared to surface-decoupled profiles.

We consider an implementation of the automatic ice detection algorithm introduced above as a promising approach for future studies. For the sake of compatibility to our previous studies (Kanitz et al., 2011) we however suggest to follow the original approach presented in the manuscript.

[Figure]

**Figure 4:** Fraction of ice-containing clouds determined using a volume depolarization threshold of 0.03. In dark blue the results for surface-decoupled clouds are shown and in cyan those for surface-coupled clouds. In orange results for Leipzig, Germany from Kanitz et al. (2011) are presented. The numbers above the plots represent the respective profile behind each data point.

*- p. 7 l. 1 - what is the slope or the metric with which cloud top is defined? This should also be specified.*
The cloud top has been derived by the highest cloud radar range gate, which was classified as cloud. The minimum detection threshold of the cloud radar was 5 times the signal to noise ratio in the co-channel. This information has been added to the manuscript at page 7, line 14-17.

*- p. 7 l. 5 - "coldest temperature" - temperatures can lower but not colder – suggest rewording here and in other locations in the text.*
Reworded.

*- p. 7 l. 6-9 - This method of using the inversion base temperature as cloud top temperature may explain some of this study's results, as the assumption becomes less valid in cases where clouds protrude into temperature inversions, which often occur concurrently with stronger mixing, not necessarily down to the surface. I think that in the context of this paper the authors might be able to make their point by defining their current "cloud top temperature" as "minimum cloud temperature", which would also be valid for cloud protruding into an inversion, and would retain the essence of INP activation temperature widely discussed in the text. Also, note that note all polar liquid-bearing clouds are capped by a temperature inversion. See for example Sedlar and Tjernström,*

*2009; https://link.springer.com/article/10.1007/s10546-009-9407-1, Sedlar et al., 2012; https://journals.ametsoc.org/view/journals/clim/25/7/jclid-11-00186.1.xml?tab_body=fulltext-display, Sotiropoulou et al., 2014; https://acp.copernicus.org/articles/14/12573/2014/, Silber et al., 2020; https://agupubs.onlinelibrary.wiley.com/doi/full/10.1029/2020GL087099*

We changed our wording from "cloud-top temperature" to "minimum cloud temperature". Additionally, as suggested by reviewer we have stated more clearly how the temperature was obtained (minimum temperature between liquid layer base and cloud top, see page 8, line 18-19).

*- p. 7 l. 12-14 - based on the fact that the authors have used the radar for this seeding cloud proximity criterion, I think that they refer here to overlying hydrometeors rather than overlying clouds. If that is correct I recommend revising the text accordingly.*
That is correct. We changed the text accordingly.

*- p. 9 l. 31 - it could be the vast majority of clouds (> 80%) contain ice but this is certainly not every cloud as currently stated in the text.*
We reworded that sentence stating that the majority of clouds contain ice.

*- Fig. 3 caption confusion - deltaT should be 5 C below -10 C and vice versa.*
Corrected.

*- p. 11 l. 13-14 - "The reasons for the increase in ice forming efficiency for low and coupled clouds in the Arctic must be caused..." - while this is likely the case, I think that the authors should tone down this sentence.*
We reworded that sentence.

*- p. 12 l.10-11 - decoupling does not necessarily mean that there is an underlying inversion, but only that the underlying layer is stable. I suggest revising the text accordingly.*
We reworded that sentence.

*- p. 12 l. 11-13 - clouds largely act to destabilize the polar atmosphere and not the opposite. Another more likely possibility is that once the marine aerosols are mixed aloft, the atmosphere becomes decoupled as a result of radiative cooling of the surrounding ice surfaces.*
We decided to make our discussion more general by stating that we do not have information about the evolution of the cloud before reaching the observation site.

*- p. 12 l. 16 - add "as" before "such"*
Done.

*- p. 12 l. 23 - define beta*
Done.

*- p. 12 l.13 - a reduction in beta is generally seen throughout the atmospheric profile regardless of the decoupling height (and sometimes increases above the decoupling height such as in the green and blue curves), so I find this argument by the authors to be rather subjective.*
The INPs abundance discussion was comprehensively reworded and made more clear. The observed increase of beta above the decoupling height is due to a cloud. This information has been added to the manuscript (page 14, line 5-6).

*- Fig. 6 - what do the normalized 0 and 2 values represent?*
0 is the surface and 2 is 2 times the decoupling height. The axes labels have been improved.

*- p. 13 l.1 - Temperature units are missing.*
Paragraph has been deleted. See answer to major remark of INPC estimation.

**Specific reply to Referee #2**

**Major comments:**

*The dataset is limited one month and half in 2017: despite the fact it must be acknowledged the considerable effort spent to collect the presented measurements, a dataset with a longer time coverage covering at least two seasons – discussed also in conjunction with a more detailed meteorological analysis - could provide more robust results. The effect of the limited dataset time coverage may have an effect on the discussed results and this should be considered.*

The reviewer rises the legitimate concern about the limitation of the dataset. Indeed we would like to analyze a longer time series, but this is what we had available for our study. We incorporated this point into the discussion of the manuscript (page 14, line 26-29).

*In the identification of ice clouds (section 2.1 Ice-containing cloud analysis), the description of the procedure applied to classify and characterize individual cloud profiles is purely qualitative, the thresholds applied to the value of the depolarization and backscattering coefficient are not mentioned indicating that the profiles have been evaluated on a subjective basis.*

We did not provide quantitative thresholds about how we separated ice and liquid clouds because of the difficulty in doing so. There have not been many studies on a quantification of the lidar volume depolarization on ice detection, as volume depolarization ratio is the superposition of molecular and particulate backscatter in the co- and cross-channels. To tackle this obstacle we used manpower to manually analyze the data set.

Therefore, we decided to describe the applied method in detail. We analyzed the complete available data set. The only periods that have been excluded from the analysis, are as described those with favorable seeding conditions (i.e., a cloud above the analyzed cloud within 1km). Combined with the fact that all data is freely available, this study should be reproducible by anybody given the description of the methodology in the article.

Nevertheless, we made a first attempt to provide a depolarization threshold on the ice detection, which is shown below. We calculated a minimum volume depolarization where the lidar should be able to detect ice. This is based on an ice water content detection threshold of $10^{-6}$ kg m$^{-3}$ (Bühl et al., 2013) which was converted into lidar extinction using the approach of Hogan et al. (2006). Using a lidar ratio of 30 sr (typical single-scattering lidar ratio of ice crystals, see, e.g., Seifert et al., 2007) we calculated the particle backscatter coefficient. The molecular backscatter coefficient at 532-nm (wavelength of the used depolarization channels) was derived using scattering theory (Hinkley, 1976) for a temperature of -10°C and air pressure of 925 hPa (ca. 700 m height). Assuming a particle depolarization ratio of ice crystals of 0.5, a minimum volume depolarization of 0.03 was found corresponding to the ice detection threshold of $10^{-6}$ kg m$^{-3}$.

This threshold was used to reproduce our study with an automatic approach. We defined a volume depolarization signal above 0.03 in four contiguous lidar range gates, but below the liquid-cloud base, as an ice layer. An ice-containing cloud profile was defined, when during half of the profile time an ice layer with volume depolarization ratio > 0.03 was detected. The results are presented in Fig. 1. While we found minor quantitative differences between the manually analyzed data set below -15°C, the main message of this manuscript remains: The occurrence of ice-containing surface-coupled cloud profiles at temperatures above -10°C is much higher compared to surface-decoupled profiles.

We consider an implementation of the automatic ice detection algorithm introduced above as a promising approach for future studies. For the sake of compatibility to our previous studies (Kanitz et al., 2011) we however suggest to follow the original approach presented in the manuscript.

[Figure]

**Figure 1:** Fraction of ice-containing clouds determined using a volume depolarization threshold of 0.03. In dark blue the results for surface-decoupled clouds are shown and in cyan those for surface-coupled clouds. In orange results for Leipzig, Germany from Kanitz et al. (2011) are presented. The numbers above the plots represent the respective profile behind each data point.

*There is no mention to the uncertainties and assumptions done in the lidar data processing (use of lidar ratios, calibration of profiles, quantification of effects like specular reflection, etc …) which are quite relevant for the presented statistics. Everything could be referred to a literature paper to clarify the data processing, but, as it stands, I am not able to find one reference for these aspect in the entire section, only one for the multiple scattering affecting the depolarization ratio.*

Similar to a comparable concern raised by reviewer 1 we agree that the issue of specular reflection is not discussed well enough in the manuscript. However, since this effect is most prominent for rather large dendrites, who form at a temperature of roughly -15°C, we think specular reflection has little influence on our findings where we found the strongest influence at a temperature above -10°C. We extended the Discussion section and elaborated the effect (see page 15, line 7 and the following**).**

*The authors uses "the cold side of the temperature inversion which is closest to the cloud-radar-derived cloud top height in the radiosonde data to defined the cloud-top temperature." It is not clear to me how large is the difference in meter between cloud top derived from the radar and the height of the radiosonde in correspondence of the cloud-top temperature. May a large difference be the result of a collocation effect which is negligible or not?*

As suggested by reviewer1 we changed the terminology from "cloud-top temperature" to "minimum cloud temperature". To derive the minimum cloud temperature we searched for the lowest temperature between liquid layer base and cloud top. This information has been added to the manuscript on page 8, line 18-19.

*In section 2.2, a scaling factor for the parameterization of DeMott et al. (2015) is derived from a single paper in literature, Gong et al. (2020), where filter samples from the Cape Verde Atmospheric Observatory were studied and INP active at temperatures above -10°C were found, which consists likely of biological material. This factor is assumed as a sort of "true" to estimate the INP concentration without any study on the sensitivity of the results to this assumption. The considered assumption may lead to large uncertainties in the retrieved INP profiles. The authors should not forget that the lidar*

*retrieval have already uncertainties and is based on assumptions the effect of which might be amplified by this further assumption in the parameterization of mineral dust.*

A comparable concern was raised by reviewer 1. We decided to remove the Section on the retrieval of the INPC together with the Figure of the INPC (Fig. 6b in the old manuscript). The low data basis makes sophisticated parametrization impossible. We tried to provide an estimate of a possible INP load but have to admit that the uncertainty of the presented approach is too large. Instead, we extended the discussion on the measured attenuated backscatter coefficient values, including the lack of parametrizations for INP in the Arctic (see page 14, line 1-16).

*3. Likewise It's unclear why the authors did not use the cloud radar measurements in the identification and filtering of cases with ice crystal precipitation. This is another points which can change the statistics collected in too subjective way, to my opinion, affecting the final results.*

As also pointed out by reviewer 1 the cloud radar Mira-35 is much more sensitive to ice detection as the lidar (the lidar has a detection threshold in IWC of about $10^{-6}$ kg m$^{-3}$, see Bühl et al. (2013)). Similar to our answer to reviewer 1, we like to point to the frequently observed low clouds during PS106 (see Griesche et al. (2020)) with a cloud base below the lowest detection range of the radar (155m above the instrument). In some cases, even the cloud top was below this height. These clouds would have been falsely classified using the cloud radar only, while these clouds are actually those closest to the surface and therefore most-likely coupled to it. Using the near-field capabilities of our polarization lidar Polly$^{XT}$ is thus a prerequisite for the presented coupling study. Another point is that we wanted to do a study that is comparable to previous studies, as the one of Kanitz et al (2011). Nevertheless, we created the same analysis of the ice-containing clouds for surface-coupled and –decoupled cases (see Fig. 2). As to be expected the amount of ice containing clouds increased, yet ice-containing surface-coupled profiles were both absolutely (i.e. the numbers of profiles) and relatively (i.e. the fraction of ice-containing clouds) more frequent. To conclude this point: we think that the challenges we would be facing including the cloud radar for the ice detection would have deformed the statistics by excluding the very near clouds, and would have made a comparison to other studies difficult. And last but not least, the observed effect (more surface-coupled ice-containing cloud profiles with a IWC down to $10^{-6}$ kg m$^{-3}$) would not be affected by radar observations. We discussed this issue on page 15, line 1-6. Additionally the results are presented in Appendix A.

[Figure]

**Figure 2:** Fraction of ice-containing clouds determined using the cloud radar for ice detection. In green the results of the complete data set is shown, in cyan for the coupled clouds and in dark blue for the decoupled clouds.

*4. For the results shown in Figure 4, the reported statistics on the number of profiles considered in the statistics poses a questions on the dependence of the results from dataset time coverage: is the number*

*of coupled ice cloud profiles much higher because these are the most recurrent cases for the investigated period of the year? This aspect must be discussed in clear way, maybe using ancillary datasets.*

Previous studies have shown that the occurrence of low level and thus likely surface-coupled clouds in the Arctic have been highest in the northern hemisphere summer. Nevertheless these are also the clouds which are one of the greatest challenges for models (Morrison et al., 2012, https://www.nature.com/articles/ngeo1332). Hence, we strongly support the request to use longer time series, to study this observed effect in more detail.

**Specific comments**

*Line 6 page 1: replace "in " with "within"*
Done.

*Line 9 page 1: the factor mentioned here in in the range 2-5, but it is not mentioned in which temperature range assumes these values. It becomes clearer from the following sentence. Please rephrase.*
We specified the temperature range.

*Lines 14-16 page 1: this sentence is not appropriate for the abstract but for the discussion section, please remove.*
The paragraph has been reworded.

*Line 8 page 6: "to date" must be at the end of the sentence.*
Done.

*Line 8 page 2: "yet" date must be at the end of the sentence.*
Done.

*Line 24 page 3: remove higher at the beginning of the line and change ": : : than do: : :" with "higher than".*
Reworded.

*Page 7: Figure caption please put "yet" at the end of the sentence of replace "an" with "a".*
'An' replaced by 'a'.

*Page 7 line 1: it is not clear to me which algorithm has been used to retrieve the cloud top height from the radar measurements, please specify.*
The cloud top has been derived by the highest cloud radar range gate, which was classified as cloud. The minimum detection threshold of the cloud radar was 5 times the signal to noise ratio in the co-channel. This information has been added to the manuscript (page 7, line 15-17).

*Page 7 line 13: the detection of cloud by the radar up to the tropopause maybe depending on the size of ice crystals and by the concurrent atmospheric attenuation. Please nuance this sentence.*
The sentence has been reworded.

*Page 7 line 17: please replace "simplified coupling algorithm" with "simplified version of the algorithm".*
Done.

*Page 10 line 8, page 11 line 1-2: do you have any reference to support your arguments?*
Instead of adding a reference for this statement, we have weakened our statement.

*Page 12 line 1: "is the strongest"or "is stronger"?*

"is stronger".

*Page 12 line 11-13: in this part of the manuscript, there is often the comparison with statistics collected in Leipzig; I am wondering if the authors can say a few words on the usage of data from one site only at the mid-latitudes to make the comparison with a more stable region like the Arctic.*
The reason for the comparison with the mid-latitude site of Leipzig is justified by the fact that we believe the free-tropospheric aerosol in the Arctic is dominated by continental long-range sources as is also the case for the free troposphere over Leipzig. This possibility has been discussed in more detail in the manuscript (page 14, line 17-25).

*Page 14 lines 14-19: in this paragraph, the reader can find the list of the limitation of the results presented in this study. These are highlighted only at the end of the manuscript as an outlook for future studies while they should be discuss also when the results are presented.*
The discussion has been completed by the respective points.

[revised manuscript text omitted]

---

## Editor Decision (ED1)

[revised manuscript text omitted]

18-4317-2018, 2018.

---

## Author Response (AR2)

**Author's response to two anonymous reviews for ACP-2020-1096**

Review #2

We thank the two referees for taking their time again to provide us with helpful comments which improve the quality of the manuscript. We have thoroughly discussed the addressed issues. Please find below our responses to the raised points (Reviewer comments are given in blue-italic).

**Report #1**

*1. I don't know how much time aerosols of different types lifted from the surface to the lower troposphere can remain airborne before settling but I doubt it is typically on the order of 10-days. The authors did not provide any reference or reasonable justification for this selection nor have they provided sensitivity test results.*

Typical periods for back-trajectory analysis in the Arctic are on the order of 5-10 days (e.g. Freud (2017, 10 days), Schmeisser (2018, 7 days), Stock (2008, 5 days)). Also a recently published model analysis showed that the aerosol transport from midlatitude sources into the Arctic can be on the order of 8 days (Zheng, 2021).

*2. Air parcels can generally speaking travel enormous distances over a period of 10-days, much longer than the great circle distance between Svalbard and Leipzig. Thus, I think that without any other contextual analysis (e.g., probability of a common geographic source), from a statistical perspective, it is not surprising that the results from Svalbard could be rather similar to those from Leipzig.*

It is well accepted that the aerosol in the free troposphere over the Arctic is dominated by aged aerosol pollution mixed with dust and wildfire smoke from all the continents around the Arctic (see Law, 2014, and the recent reviews of Abbatt, 2019 and Willis, 2018). However, we acknowledge the hint that our analysis lacked context. Therefore we added a comparison of a similar aerosol source analysis published in Radenz (2021a) for Krauthausen, Germany, to the manuscript. The distribution of the possible source regions above an arrival height of 3 km show a comparable pattern to what we have found for the analyzed period in the Arctic. One exception is the importance of 'barren', which contributes in the Arctic only for trajectories arriving above 6 km altitude.

*3. I understand that a reception height of 2 km is widely used in the literature. That does not suggest that it serves as a reasonable assumption in every case, and specifically in this case, where most clouds are much closer to the surface, and hence, an altitude of 2 km is not representative and lacks context.*

The reception height is used along the trajectory to identify time periods when the air parcel was in the vicinity of the surface and hence the air mass was possibly influenced by aerosol sources on the ground. The height distribution of the clouds at the trajectory destination should play a minor role in this matter. As an alternative to a fixed reception height, the model-derived local mixing depth can be applied. Yet, models often do not consider the residual layer and hence they tend to underestimate the mixing layer depth (Vivone, 2021), even in high resolution models like WRF (Banks, 2015). This is why the approach of Radenz et al. (2021) focuses on fixed reception height. Nevertheless, we performed the same analysis as presented in the manuscript using the model-derived mixing layer depth as reception height. The respective results are shown in Fig. 1. The largest contribution of 'snow/ice' as a possible source region has shifted from trajectories reaching the position of Polarstern at heights around 3 km altitudes to below 2 km. Also, the contribution of 'savanna/shrubland' has become more frequent in all heights. Overall the attribution of possible source regions to the aerosol burden at the analyzed location qualitatively still compares well to what has been published Radenz (2021a) for a campaign performed in Krauthausen, Germany, confirming continental conditions in the free troposphere over RV Polarstern during the time period investigated in the frame of our study.

[Figure]

*Figure 1. Fraction of residence time of air parcels arriving at heights between 0 and 10 km below the model-derived local mixed layer depth based on a FLEXPART 10 days back-trajectory analysis.*

*4. The authors did not elaborate on this issue, but HYSPLIT (assuming that this is the model used by the authors) often tends to continue the back-trajectory calculations even after the parcel reaches the surface. Did the authors remove such cases in which the parcel apparently reached its source?*

Our analysis was actually based on FLEXPART trajectories. FLEXPART uses a more sophisticated treatment of turbulence (Stohl, 2005) and hence the particles do not 'stick' to the surface for longer periods, as they can do in HYSPLIT. Terminating the particles at surface contact would then imply the disappearance of air parcels and hence a violation of

continuity. Additionally, contacts with the surface between the output time steps cannot be diagnosed afterwards.

However, a sensitivity analysis with the HYSPLIT variant of the air mass source estimate revealed no significant difference in the campaign averaged residence times.

*5. In general, a single paragraph is definitely insufficient to introduce, describe, and conclude from a new analysis of the data, but on the other hand, I doubt how much further the authors would like to elaborate on this analysis because it may cause the manuscript to lose some of its focus.*

We thank the reviewer for pointing out the weaknesses of our analysis and the way it was presented. The respective paragraph was reworded and complemented by missing information addressed in this review (see page 15, line 24 and the following in the diff-version).

*Minor comments:*

*1. Recommended swapping sec. 4.1 with 4.2. It seems to me like sec. 4.2 better fits the beginning of the Discussion right after the results are presented, while sec. 4.1 could provide a smoother transition to the Summary & Conclusions.*

We swapped the respective sections.

*2. P. 15 l. 2 - Suggest changing "lowest detection limit" to "lowest radar range gate" because detection limit often insinuates intensity limitation.*

Done

*3. Caption of figure 8: perception --> reception.*

Done

**Report #2**

*After a detailed reading of the authors' comments and of the updated version of the manuscript, I am still concerned about the approach adopted by the authors. I must acknowledge that the authors extended the presented data analysis to support their conclusions. Nevertheless, I regret to note there are still not fully transparent aspects and not obvious choices in the data analysis which does not allow the presented analysis to be complete and to improve the accuracy of results.*

*My concerns can be summarized in the two following major points.*

*The authors state that they initially "did not provide quantitative  thresholds about how we separated ice and liquid clouds because of the  challenges in their estimation. As a consequence their preferred to use  manpower to manually analyze the data set and decide where ice is or  not." Nevertheless, In their response, the authors also provides an approach fundamentally based on the definition of a threshold on lidar  volume depolarization ratio for the detection of ice-containing  clouds.The comparison between the "manual" data selection and those  based on the automatic data classification, independently on the  difficulties intrinsic to the definition of credible thresholds for the automatic algorithm, opens the way to the following thought: considering  that with the automatic selection the difference in the percentage of  surface-coupled and decoupled clouds is lower compared to the manual  analysis (i.e a factor of about 1.5-4 vs a factor of 2-6 on average in  between 0°C and -10°C) can we consider this difference as the results of  the level of subjectivity of the analysis? Or is this an indication  that the analysis is largely affected by the irreducible uncertainties  due to the assumptions done in the retrieval of lidar products? Assuming  the error bars in Fig.5 are a good estimation of statistical  uncertainty according to Seifert et al. (2010), the variability between  the "manual" and "automatic" data processing may be representative of a  bias which may affect your manual approach. Although the results  demonstrates that majority of clouds in the height corresponding to the  interval 0°C and -10°C are surface-coupled, the quantification of their  fraction must be as accurate as possible and potential systematic effects, such as those due to a manual data analysis, should be  discussed in the manuscript.*

In lidar research, a manual analysis is by far the most accurate approach as this allows the possibility to check the reliability of the basic, fundamental signal profiles as well as the retrieval products by your experienced eye (one can do that back and forth several times in the data analysis). This is well known and accepted in the lidar community. There are numerous examples available, in which manual analyses reveal deficiencies in automatic retrievals. Just to name some studies of our group, related to evaluation of measurements of the spaceborne lidar CALIOP: Wandinger et al. (2012) and Kanitz et al. (2014). All the attempts to introduce automated data analysis schemes (as for example in the case of EARLINET, D'Amico et al., 2015) were motivated by the fact that more and more lidars deliver continuous observations. But all these products obtained with automated analysis schemes have to be convincingly compared with manually analyzed products, before the (lidar) community trusts them. So, there is no doubt: If the chance is given to apply best knowledge in a manual analysis, there is no better alternative. In our case, the chance was given, as the temporal extent of the dataset allowed us to do this with justifiable efforts, and given the unprecedented measurement capabilities (not available so far for observations in the marine environment of the summer Arctic). In our analysis we thus followed the well-established methodology derived and refined by Ansmann et al. (2009), Seifert et al. (2010, 2011, 2015) and Kanitz et al. (2011).

We once more take the opportunity of this reply letter to highlight to Reviewer #2 and the Editor that our studies are the very first approach of a detailed analysis of the structure of extremely low-level Arctic mixed-phase clouds and their response to surface coupling. This was - to our knowledge - not done before, also because of technical caveats of

measurement systems deployed so far in the Arctic (and in the marginal sea ice zone over the open ocean). To our knowledge, there was no polarization-sensitive lidar deployed so far in the marine Arctic, which can provide a liquid/ice separation at heights starting as close as 50 m above ground. Please consider, meanwhile new studies are underway which go into a similar direction with respect to surface coupling effects on ice formation. E.g., Radenz et al. (2021b) found surface-coupling effects on heterogeneous freezing in clouds observed in Southern Chile. They, however, did not have to deal with the very low cloud layers, as they are subject to our Arctic study discussed here. These low clouds provide a great challenge to lidar observations, as they frequently occur at heights within the incomplete laser-beam receiver-field-of-view overlap, even in the case of a lidar with near field capabilities like the PollyXT.

The reliability of an automated data analysis is strongly constrained by an accurate base height of the liquid dominated layer. If this base is set too high, multiple scattering present in the liquid dominated layer might be classified as ice occurrence. A base height located too low might omit depolarization signals from ice crystals below the cloud and hence cause a misclassification of ice containing clouds as liquid clouds. It requires further investigations to carefully consider such effects in automatic retrievals. Especially in the absence of appropriate radar measurements.

We recommend to not elaborate further on the automatic retrieval of our statistics within this manuscript. The approach shown in the reply letter #1 was supposed to briefly evaluate any systematic effects on the approach we present in the manuscript. Even though the automatic retrieval appears of certain value and triggers certain interests, issues resulting from multiple-scattering effects cannot be neglected and need further investigation, which is outside the scope of this study. The application of thresholds in the automated approach combined with the frequently occurring low-level clouds is likely contributing to the difference between the statistics derived using the automated and the manual analysis (see: Figure 4 in reply letter #1). In addition, it is not possible to separate ice-containing from liquid-only clouds below a height of 80m, as the minimum height of the lidar signal is 50m and the minimum ice layer thickness used to identify ice in the depolarization profile was set to 30m. The difference in frequency of occurrence of ice-containing clouds between the manual and the automated approach for clouds with a cloud minimum temperature above -5 °C is likely a consequence. Moreover, this method underestimates pure ice clouds which are lacking a liquid layer and clouds where the ice is located above the liquid layer. This is likely a cause of the differences in ice containing clouds below -10°C.

Thus, evaluating the differences between the automatic and the manual approaches is to our opinion no suitable approach for estimating the level of subjectivity of the analysis. The presented automatic approach comes with its own deficiencies which cause further ambiguities. We hope and wish that our study encourages future studies which are based on enhanced instrumental and thus methodological capabilities.

A longer-lasting dataset would have been of advantage for our study. However, on the one hand this dataset is what we had at hand for our study. It is the first one ever with the PollyXT near-field capabilities plus collocated cloud radar observations. On the other hand, the rather short duration also opened up the possibility of the manual best-knowledge analysis. In our opinion, we provide a thorough discussion of the possibilities for future refinements which can be applied to future studies, hopefully covering longer time periods.

*2. The authors removed from the manuscript the analysis on the estimation of INP concentration and added a new analysis to demonstrate the hypothesis that the aerosol source acting as INPs in Arctic are the similar to those of a continental site, i.e. Leipzig, which is used as a comparison term in the data analysis.*

*First of all it is not clear to me why, using a multi-wavelength Raman lidar, aerosol extinction profiles in clear sky, presented in the updated manuscript version, and consequently, lidar ratio profiles are not retrieved from Raman channels. The capability of a multi-wavelength Raman lidar are fully neglected with consequent increase of the uncertainties in the retrieved products. Assuming a constant value of the lidar ratio may be considered acceptable only for ice cloud, although also I that case the variability of lidar ratio could affect at smaller extent the selected thresholds. The retrieval of the aerosol lidar ratio from Raman lidar measurements from could definitely, coupled with the particle depolarization ratio and air mass back-trajectories, clarify the role of different aerosol types involved the ice clouds formation (several example from literature can be provided, but I am sure the authors knows all of them very well). This could also avoid to include the comparison with Leipzig in support of the authors interpretation of a major role for the continental aerosol in the ice cloud formation in the Arctic and increasing the credibility of the presented analysis, which appears still too speculative.*

An in-depth lidar analysis using the full range of the capabilities a Raman lidar offers could have been helpful for a detailed characterization of the lofted aerosol layers. Yet, for our contrasting analysis we need robust (i.e. not influenced by overlap problems) quantities, which are the backscatter coefficient and the depolarization ratio. Both quantities are determined from signal ratios so that incomplete overlap effects cancel out (in case of a well adjusted lidar). We continuously ensured the good performance of the lidar during our manual data analysis, as well as already on-board of RV Polarstern (in presence of the first author of this study, Hannes Griesche) during acquisition of the measurements.

Backscatter coefficient and depolarization ratio alone already allow the identification of dust. Extinction coefficients and lidar ratios are useful to have but it is usually not sufficient to unambiguously determine the aerosol type. In our study, the focus is on the INP efficiency for aerosol in the lowest heights, where only dust and biological particles are of importance (e.g. Abbatt., 2019; Willis, 2018). In these low altitudes, extinction and lidar ratio information are of no advantage. Additionally, since the analyzed cruise was conducted in the Arctic summer in the Arctic ocean we were measuring under continuous daylight conditions. The high background signal prevented us from performing a multi-wavelength Raman analysis of aerosol extinction and lidar ratios.

*I want also to add that aerosol backscatter profiles in Fig. 7 must be shown over a longer vertical range to ensure the reader can have a clear idea of the calibration accuracy of lidar profiles.*

We have developed and applied sophisticated methods to accurately calibrate lidar signals and signal ratios to obtain quality assured backscatter coefficients. The applied methods, such as Rayleigh fitting, determining the lidar constant to get even get backscatter profiles below cloud decks, when Rayleigh calibration in clear skies in the upper troposphere is not possible, are described in Baars (2016), Hofer (2017), Haarig (2017) and Jimenez (2020).

There is thus no need to show higher-reaching profiles, as it would distract the reader from the conditions in the vicinity of the coupling height.

*About the presented investigation of the aerosol sources using FLEXPART model, if it is true that marine fraction decrease with height, above 2 km, it remains the major source at all levels, while the other aerosol types slightly increases with the height, except for "grass cropland" aerosol which has a bit larger increase. I do not see any reference to support the authors' statement of similarity with the aerosol composition typical for the Leipzig site, which is not a marine site. For this part, the analysis looks carried in out in hurry and must be deeper.*

Similar to a comparable comment done by Reviewer 1 we like to point out that it is well accepted that the aerosol in the free troposphere over the Arctic is dominated by aged aerosol pollution mixed with dust and wildfire smoke from all the continents around the Arctic (see Law et al., 2014, and the recent reviews of Abbatt, 2019, and Willis, 2018).

Nevertheless we wanted to assess if our analysis is consistent with the literature. The results of the presented aerosol source analysis above 2 km height is comparable to a analysis of a multi-week campaign conducted in Krauthausen, Germany, presented in Radenz (2021a).

*At this point, although the statistics presented in the manuscript on the surface-coupled and decoupled ice-containing clouds are interesting, I am not sure if the content of the manuscript is sufficient for publication on ACP. Major revisions are still required and if points above cannot be fulfilled in a short time by the authors, I'd suggest to re-submit the manuscript once data are more consolidated.*

We see strong reasoning for considering this manuscript for publication in ACP:

- For the first time, we have analyzed Arctic lidar observations with focus on the relationship between the phase partitioning and surface coupling of the typically low-level Arctic cloud systems over the open ocean. This was technically just not possible before.
- We have used well established, well proven, extensively documented, reproducible methods accompanied by a careful data analysis.
- We show for the first time that heterogeneous ice formation in Arctic mixed-phase clouds depends, besides the minimum cloud temperature, on the liquid layer base height and the surface-coupling state.
- For the first time we showed that in the Arctic summer the thermodynamic linkage between the cloud and the surface increases the frequency of occurrence of surface-coupled ice-containing clouds by a factor of up to 3 compared to surface-decoupled clouds above a cloud minimum temperature of -10°C.
- For the first time, we found that the likelihood of occurrence of an ice-containing cloud is up to 6 times higher at a cloud minimum temperature above -10°C if the cloud is coupled to the surface.

- We acknowledge by means of an extensive discussion the potential and needs for future, extended studies. By doing so, we actively support the advancement of science.

We hope all these arguments are convincing enough to accept the paper.

Abbatt et al., ACP, 2019, https://doi.org/10.5194/acp-19-2527-2019

Ansmann et al., JGR, 2009, https://doi.org/10.1029/2008JD011659

Baars et al.,ACP, 2016, https://doi.org/10.5194/acp-16-5111-2016

Banks, Boundary-Layer Meteorol, 2015, https://doi.org/10.1007/s10546-015-0056-2

D'Amico et al., AMT, 2015, https://doi.org/10.5194/amt-8-4891-2015

Freud et al., ACP, 2017, https://doi.org/10.5194/acp-17-8101-2017

Haarig et al., ACP, 2017, https://doi.org/10.5194/acp-17-10767-2017

Hofer et al., ACP, 2017, https://doi.org/10.5194/acp-17-14559-2017

Jimenez et al., ACP, 2020, https://doi.org/10.5194/acp-20-15247-2020

Kanitz et al., GRL, 2011, https://doi.org/10.1029/2011GL048532

Kanitz et al., AMT, 2014, https://doi.org/10.5194/amt-7-2061-2014

Law et al., BAMS, 2014, https://doi.org/10.1175/BAMS-D-13-00017.1

Radenz et al., ACP, 2021a, https://doi.org/10.5194/acp-21-3015-2021

Radenz et al., ACPD, 2021b, https://doi.org/10.5194/acp-2021-360

Schmeisser et al., ACP, 2018, https://doi.org/10.5194/acp-18-11599-2018

Seifert et al., JGR, 2010, https://doi.org/10.1029/2009JD013222

Seifert et al., JGR, 2011, https://doi.org/10.1029/2011JD015702

Seifert et al., GRL, 2015, https://doi.org/10.1002/2015GL064068

Stock et al., Atmos Environ, 2008, https://doi.org/10.1016/j.atmosenv.2011.06.051

Vivone, ACP, 2021, https://doi.org/10.5194/acp-21-4249-2021

Willis et al., Rev. Geophys, 2018, https://doi.org/10.1029/2018RG000602

Wandinger et al., GRL, 2010, https://doi.org/10.1029/2010GL042815

Zheng et al., 2021, JGR, https://doi.org/10.1029/2020JD033811